# Avoidance of hydrogen sulfide is modulated by external and internal states in *Caenorhabditis elegans*

Longjun Pu[1,2,3], Lina Zhao[1,2,3], Jing Wang[1,2,3], Clementine Deleuze[4], Lars Nilsson[1,2,3], Johan Henriksson[1,5,6], Patrick Laurent[4]*, Changchun Chen[1,2,3]*

[1]Department of Molecular Biology, Umeå University, Umeå, Sweden; [2]Umeå Centre for Molecular Medicine, Umeå University, Umeå, Sweden; [3]Wallenberg Centre for Molecular Medicine, Umeå University, Umeå, Sweden; [4]Laboratory of Neurophysiology, ULB Neuroscience Institute (UNI), Université Libre de Bruxelles (ULB), Brussels, Belgium; [5]The Laboratory for Molecular Infection Medicine Sweden (MIMS), Umeå University, Umeå, Sweden; [6]Integrated Science Lab (Icelab), Umeå University, Umeå, Sweden

*For correspondence:
patrick.laurent@ulb.be (PL);
changchun.chen@umu.se (CC)

Competing interest: The authors declare that no competing interests exist.

## eLife Assessment

These **valuable** studies explore the consequences of exposure to the toxin hydrogen sulfide (H2S) on the behavior and physiology of *C. elegans*. The work finds that behavioral changes evoked by H2S exposure are modulated by several regulatory pathways known to influence chemosensory-evoked locomotor behavior, but there is **incomplete** data to support the authors' claim of comprehensive mechanistic insight into the consequences of H2S exposure. Nevertheless, the findings may be informative for those studying organismal stress responses and the effects of mitochondrial ROS on behavior and physiology.

**Abstract** Hydrogen sulfide ($H_2S$) acts as an energy source, a toxin, and a gasotransmitter across diverse biological contexts. We use the robust locomotory responses of *Caenorhabditis elegans* to high levels of $H_2S$ to elucidate the molecular mechanisms underlying its acute and adaptive responses. We find that the $H_2S$-evoked behavioral response is shaped by multiple environmental factors including oxygen ($O_2$) levels and nutritional state and is modulated by various pathways such as insulin, TGF-β, and HIF-1 signaling, as well as by input from $O_2$-sensing neurons. Prolonged exposure to $H_2S$ activates HIF-1 signaling, leading to the upregulation of stress-responsive genes, including those involved in $H_2S$ detoxification. This promotes an adaptive state in which locomotory speed is reduced in $H_2S$, while responsiveness to other stimuli is preserved. In mutants deficient in HIF-1 signaling, iron storage, and detoxification mechanisms, animals display a robust initial response but rapidly enter a sleep-like behavior characterized by reduced mobility and diminished responsiveness to subsequent sensory stimuli. Furthermore, while acute production of mitochondria-derived reactive $O_2$ species (ROS) appears to initiate the avoidance response to $H_2S$, persistently high ROS promotes an adaptive state, likely by activating various stress-response pathways, without substantially compromising cellular $H_2S$ detoxification capacity. Taken together, our study provides comprehensive molecular insights into the mechanisms through which *C. elegans* modulates and adapts its response to $H_2S$ exposure.

## Introduction

During the late Proterozoic era, the rise in oxygen ($O_2$) levels eliminated hydrogen sulfide ($H_2S$) from most habitats. Yet, low $O_2$ levels, along with high concentrations of hydrogen sulfide ($H_2S$) and carbon dioxide ($CO_2$), can persist in enclosed environments where bacteria actively decompose organic matter (*Olson and Straub, 2016*). $H_2S$ can easily permeate biological membranes and interfere with various cellular processes. One of the most detrimental effects of $H_2S$ is the disruption of cellular respiration by remodeling the mitochondrial electron transport chain and inhibiting cytochrome *c* oxidase (COX; *Cooper and Brown, 2008*; *Khan et al., 1990*; *Nicholls and Kim, 1982*; *Romanelli-Cedrez et al., 2024*).

The detection and response to chemical signals are crucial for various organisms to interact effectively with their environment. Animals living in $H_2S$-rich environments have evolved mechanisms to either avoid or adapt to these conditions, conferring them survival advantages. Species that encounter periodic increases in $H_2S$ often exhibit avoidance behaviors. For example, benthic species undertake daily vertical migrations between sulfidic and non-sulfidic waters (*Abel et al., 1987*; *Salvanes et al., 2011*). Certain metazoan species have seized the ecological opportunities presented by $H_2S$-rich environments through the development of cellular adaptations. Live-bearing fish species, for instance, have evolved COX subunits that are more resistant to $H_2S$ and more efficient at $H_2S$ detoxification (*Kelley et al., 2016*; *Pfenninger et al., 2014*).

Free-living nematodes feed on bacteria that thrive in decaying organic matter, such as compost. This complex and dynamic environment, where species like *Caenorhabditis elegans* are commonly found, may exhibit significant fluctuations in $H_2S$ levels over short distances (*Adams et al., 1979*; *Budde and Roth, 2011*; *Morra and Dick, 1991*; *Patange et al., 2025*; *Rodriguez-Kabana et al., 1965*; *Romanelli-Cedrez et al., 2024*). Evidence suggests that *C. elegans* has evolved tolerance to prolonged $H_2S$ exposure by reprogramming gene expression through $H_2S$-induced stabilization of the hypoxia-inducible factor HIF-1 (*Budde and Roth, 2010*; *Budde and Roth, 2011*; *Horsman et al., 2019*; *Ma et al., 2012*; *Miller et al., 2011*; *Powell-Coffman, 2010*; *Topalidou and Miller, 2017*), and by developing an $H_2S$-resistant electron transport chain (*Romanelli-Cedrez et al., 2024*). In particular, low concentrations of $H_2S$ are not only well tolerated but can also be beneficial. For instance, exposure to 50 ppm $H_2S$ has been shown to extend the lifespan and enhance thermotolerance (*Fawcett et al., 2015*; *Miller and Roth, 2007*). Moreover, endogenous $H_2S$ production in *C. elegans* modulates its interactions with actinobacteria (*Patange et al., 2025*). However, exposure to concentrations exceeding 150 ppm proved lethal (*Budde and Roth, 2010*; *Horsman et al., 2019*).

In addition to long-term adaptive mechanisms, *C. elegans* exhibits an acute behavioral response to $H_2S$ exposure (*Budde and Roth, 2011*). In this study, we investigated whether and how the nematode *C. elegans* acutely avoids environments rich in $H_2S$, how it adapts at the molecular and physiological levels to $H_2S$ exposure, and how these adaptations alter the avoidance response.

## Results

### *C. elegans* increases locomotory activity in response to acute $H_2S$ exposure

In response to aversive cues, *C. elegans* initiates a pirouette followed by an increase of its locomotory speed to escape the noxious stimuli. When acutely exposed to 150 ppm $H_2S$, the laboratory N2 strain exhibited an increased locomotory speed and enhanced turning behavior, while generally decreasing the frequency of reversals, as a strategy to escape the noxious stimuli (*Figure 1A–C*, *Figure 1—video 1*). The maximum speed was reached after 6–8 min in $H_2S$ and returned to the baseline over a period of approximately 1 hr (*Figure 1D*). The reduced locomotory speed after 30 min in 150 ppm $H_2S$ was reversible when these animals were immediately challenged with another stimulus, such as near-UV light (*Figure 1E*), suggesting that neuromuscular function remains preserved. The magnitude and dynamics of the $H_2S$-induced avoidance responses were concentration-dependent. Under our assay conditions, 50 ppm $H_2S$ did not elicit an acute speed increase, and responses to 75 ppm were variable. In contrast, exposure to 150 ppm $H_2S$ consistently triggered a robust and reproducible escape response (*Figure 1F*), consistent with earlier reports that $H_2S$ at 50 ppm exerts beneficial effects, while $H_2S$ at 150 ppm is toxic to *C. elegans* (*Budde and Roth, 2010*; *Fawcett et al., 2015*; *Horsman et al., 2019*; *Miller and Roth, 2007*). To investigate whether low levels of $H_2S$ might be preferable or even

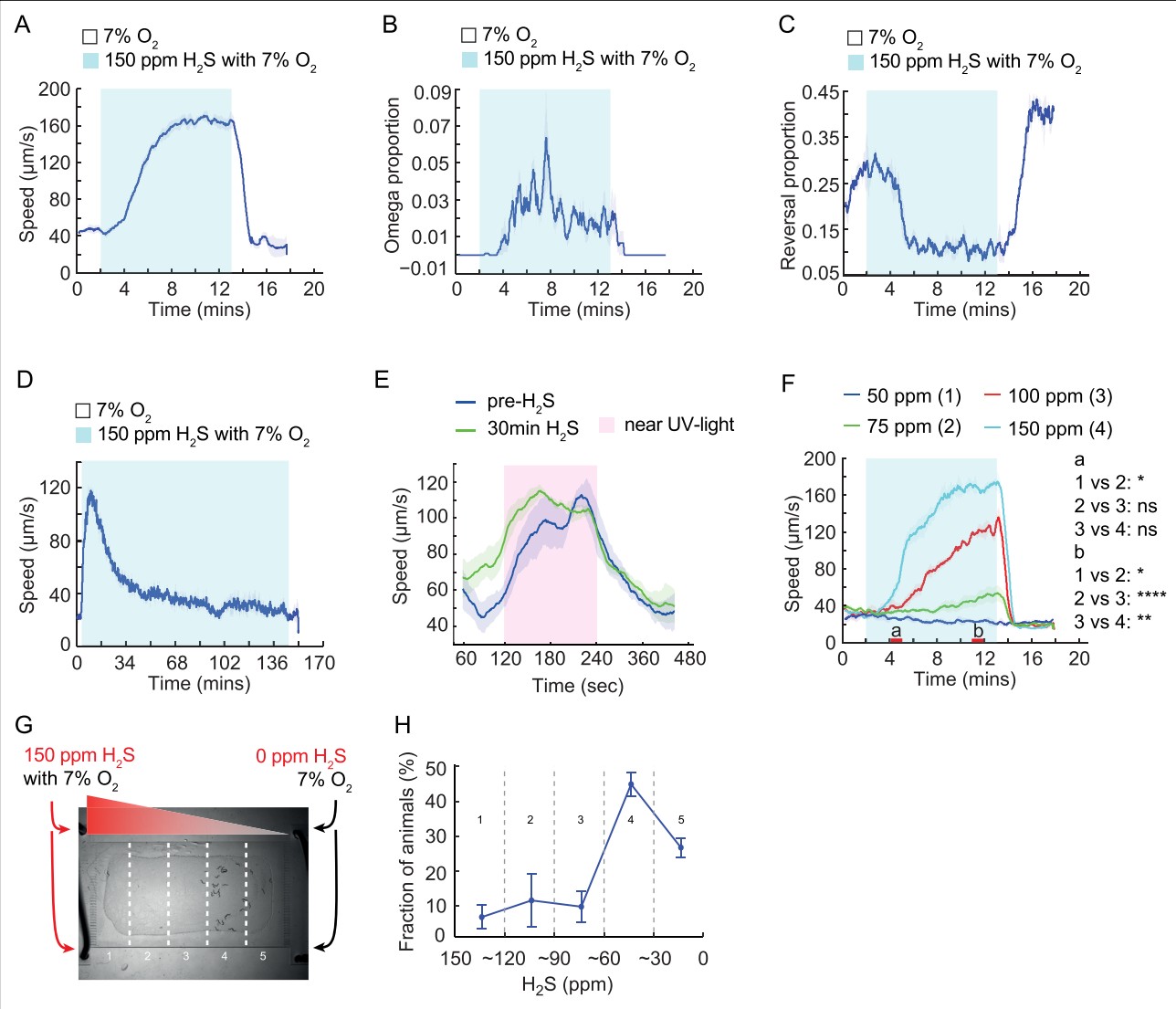

**Figure 1.** Acute locomotory responses of *C. elegans* to hydrogen sulfide. (**A**) Changes in the locomotory speed of WT animals (N2 laboratory strain) evoked by a switch from 7% $O_2$ to 150 ppm $H_2S$ balanced with 7% $O_2$, followed by a return to 7% $O_2$. (**B**) Proportion of WT animals undergoing reorientation movements (omega turns) during a switch from 7% $O_2$ to 150 ppm $H_2S$ balanced with 7% $O_2$, followed by a return to 7% $O_2$. (**C**) Fraction of WT animals doing backward locomotion (reversals) during a switch from 7% $O_2$ to 150 ppm $H_2S$ balanced with 7% $O_2$, followed by a return to 7% $O_2$. (**D**) Locomotory speed of WT was recorded for 2 min at 7% $O_2$, followed by 148 min at 150 ppm $H_2S$ balanced with 7% $O_2$, and then at 7% $O_2$. (**E**) Locomotory speed of WT animals in response to near-UV light (435 nm, 0.7 mW/mm²) exposure, before and directly after 30-min preincubation in 150 ppm $H_2S$ balanced with 7% $O_2$. Locomotory activity was recorded for 2 min before, during, and after UV light exposure. (**F**) Locomotory speed of WT animals to different concentrations of $H_2S$ balanced with 7% $O_2$. Red bars on the x-axis represent two intervals (4–5 min and 11–12 min, labeled a and b, respectively) used for statistical analysis. **** = $p < 0.0001$, ** = $p < 0.01$, * = $p < 0.05$, ns = not significant, Mann–Whitney U test. (**G**) Distribution of WT animals in a microfluidic device after 25 min of aerotaxis. The gas inputs and the five chamber sections used for scoring are indicated. (**H**) Distribution of WT animals in a gradient of $H_2S$ (150 ppm to 0 ppm). The bins correspond to different sections of the microfluidic chamber described in (**G**). N = 4 aerotaxis assays.

The online version of this article includes the following video for figure 1:

**Figure 1—video 1.** The locomotory response of WT animals to $H_2S$.

https://elifesciences.org/articles/92964/figures#fig1video1

attractive to the animals, we exposed wild-type animals to a linear gradient of $H_2S$ (*Figure 1G*; See methods for details; *Bretscher et al., 2008*; *Chang et al., 2006*; *Gray et al., 2004*). Animals were allowed to move freely on a thin layer of bacteria within a microfluidic chamber containing an $H_2S$ gradient ranging from 150 ppm at one end to 0 ppm at the other. As expected, *C. elegans* robustly

avoids 150 ppm $H_2S$ but does not accumulate at the extreme end of 0 ppm (*Figure 1G and H*). The highest proportion of animals was found in the region with approximately 40 ppm $H_2S$. These data suggest that $H_2S$ acts as a potent repellent for *C. elegans* at high concentrations but as an attractant at low levels.

## Candidate gene screen identifies signaling pathways required for $H_2S$-evoked locomotion

To investigate the molecular mechanisms underlying the $H_2S$-evoked locomotory response in *C. elegans*, we conducted a candidate gene screen for mutants that failed to increase their locomotory speed upon exposure to an acute rise of $H_2S$ level to 150 ppm. Our analysis prioritized genes that are either known to be, or potentially involved in, sensory responses to gaseous stimuli in both *C. elegans* and mammals (*Supplementary file 1*). We began by examining mutants affecting globins, guanylate cyclases, and cyclic nucleotide-gated (CNG) channels, which are involved in $O_2$ or $CO_2$ sensing in *C. elegans*, as well as wild isolates and other previously characterized mutants with defective responses to gas stimuli (*Supplementary file 1*, *Bretscher et al., 2008*; *Gray et al., 2004*; *Hallem and Sternberg, 2008*; *Persson et al., 2009*). Given the essential role of $K^+$ channels in acute hypoxia sensing in mammals, we also screened all potassium ($K^+$) channel mutants (*Gao et al., 2017*; *Weir et al., 2005*). To explore the neuronal components involved in $H_2S$-evoked avoidance, we assayed mutants with defects in ciliogenesis or cilia-mediated signaling, as well as animals deficient in neurotransmission involving classical neurotransmitters, neuropeptides, or biogenic amines. Finally, considering that mitochondrial function is central to $H_2S$ detoxification, we included mitochondrial electron transport chain (ETC) mutants and key factors implicated in $H_2S$ and superoxide clearance. Together, these functionally diverse categories allowed us to survey a wide spectrum of potential mechanisms contributing to the behavioral response to $H_2S$.

In the screen, a collection of mutants displayed reduced speed response, whereas only a few exhibited a lack of omega-turn and/or reversal responses to $H_2S$ (*Supplementary file 1*). The reversal inhibition upon $H_2S$ exposure and its rebound following $H_2S$ withdrawal appeared to vary substantially across experiments for N2, suggesting that differences in population density, food availability, and animals' physiological state prior to the assay may matter. Nevertheless, our screen revealed that signaling pathways, including cGMP, insulin, and TGF-β signaling, contributed to $H_2S$-evoked avoidance (*Supplementary file 1*). A naturally occurring variant of the neuropeptide receptor gene *npr-1*, previously implicated in the modulation of $O_2$ and $CO_2$ responses, also affected locomotory speed in $H_2S$ (*Supplementary file 1*). In addition, mitochondrial components, including those involved in respiration and $H_2S$ detoxification, played critical roles in triggering and maintaining escape behavior in $H_2S$. In contrast, we did not find evidence that globins, $K^+$ channels, or biogenic amine signaling contributed significantly to the behavioral avoidance to $H_2S$ (*Supplementary file 1*).

## cGMP signaling in ASJ cilia contributes to $H_2S$ avoidance

One group of mutants exhibiting reduced locomotory speed in $H_2S$ had defects in ciliogenesis (*Figure 2A–C*). This group included *daf-19* mutants lacking cilia, *dyf-3* mutants with reduced cilia length, and *dyf-7* mutants in which ciliogenesis occurs but the cilia are not anchored to their sensilla and are not exposed to the external environment (*Heiman and Shaham, 2009*; *Murayama et al., 2005*; *Starich et al., 1995*; *Swoboda et al., 2000*). These findings suggest that ciliogenesis and exposure of cilia to the external environment are essential for animals to escape $H_2S$ exposure. In addition, these mutants exhibited consistently high reversal rate, likely associated with their reduced locomotory speed (*Figure 2—figure supplement 1A–F*). Cilia-mediated sensory responses often involve the activation of guanylate cyclases and the opening of cGMP-gated channels (*Ferkey et al., 2021*). We observed significantly attenuated locomotory speed in response to $H_2S$ in *tax-4* or *tax-2* mutants, which lack key subunits of cilia-enriched cGMP-gated channels (*Figure 2D and E*). In a comprehensive survey of all guanylate cyclase mutants, the *daf-11* strain emerged as the only mutant showing speed response defects in the avoidance of $H_2S$ (*Figure 2D*, *Supplementary file 1*). Similar to the cilia mutants, animals deficient in *daf-11* had low speed, attenuated omega turns, and constantly high reversal rate (*Figure 2D*, *Figure 2—figure supplement 1G, H*). Selective expression of *daf-11* in ASJ neurons, but not in other neurons, partially restored the speed response to acute $H_2S$ (*Figure 2F*, *Figure 2—figure supplement 2A–D*). In addition, the $H_2S$ response defect of *tax-4* mutants was also

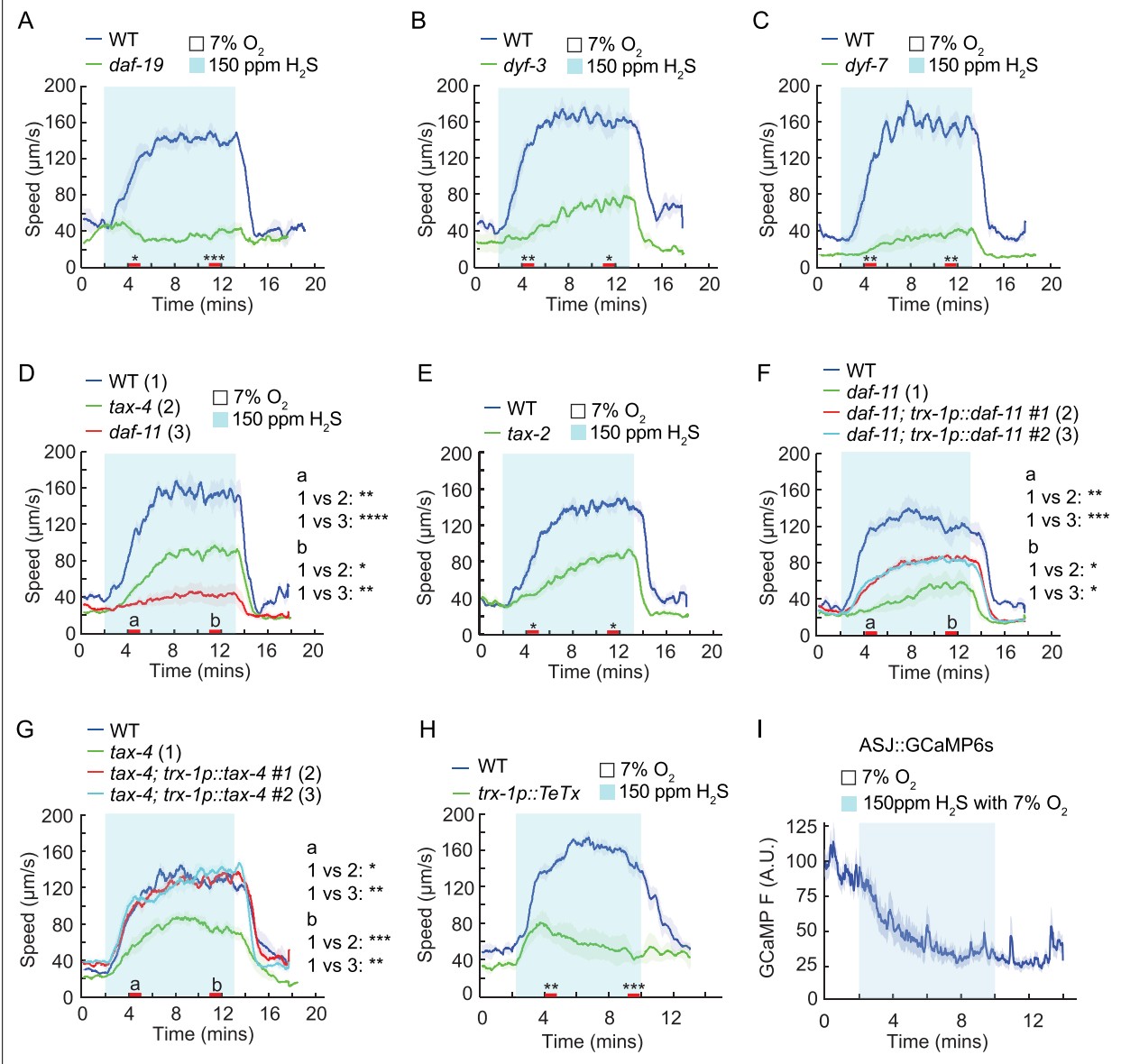

**Figure 2.** Neurosecretion from ASJ neurons contributes to H$_2$S avoidance. (**A–H**) Locomotory speed responses to a switch from 7% O$_2$ to 150 ppm H$_2$S balanced with 7% O$_2$ for animals of the indicated genotype: WT and *daf-19(m86)*(**A**); WT and *dyf-3(m185)*(**B**); WT and *dyf-7(m539)*(**C**); WT, *daf-11(m47)*, and *tax-4(p678)*(**D**); WT and *tax-2(p691)* (**E**); WT, *daf-11(m47)*, and transgenic *daf-11(m47)* expressing *daf-11* genomic DNA in ASJ neurons (two independent lines) (**F**); WT, *tax-4(p678)*, and transgenic *tax-4(p678)* expressing *tax-4* cDNA in ASJ neurons (two independent lines) (**G**); WT and transgenic WT expressing the catalytic domain of the *tetanus* toxin (TeTx) in ASJ neurons (**H**). For the comparison of acute locomotory speed responses between strains, red bars on the x-axis represent two intervals (4–5 min and 11–12 min, labeled a and b, respectively, in panels D, F, and G) used for statistical analysis. **** = p < 0.0001, *** = p < 0.001, ** = p < 0.01, * = p < 0.05, ns = not significant, Mann–Whitney U test. (**I**) Calcium transients evoked in ASJ neurons of WT animals in response to a switch from 7% O$_2$ to 150 ppm H$_2$S balanced with 7% O$_2$ for animals (N=42). The slow decline in GCaMP6s fluorescence is due to photobleaching.

The online version of this article includes the following figure supplement(s) for figure 2:

**Figure supplement 1.** Acute avoidance response to H$_2$S is regulated by cGMP signaling.

**Figure supplement 2.** Acute response to H$_2$S is regulated by *daf-11* signaling in ASJ neurons.

rescued by selective expression of its cDNA in ASJ neurons (*Figure 2G*), suggesting that the receptor guanylate cyclase DAF-11 and the CNG channel TAX-4 act at least partially in ASJ neurons to promote acute avoidance to H$_2$S. To validate the role of ASJ neurons in H$_2$S responses, we blocked neurotransmission from ASJ using the catalytic domain of *Tetanus Toxin (TeTx)*, which impairs neurosecretion

by specifically cleaving synaptobrevin (*Schiavo et al., 1992*). Cell-specific expression of *TeTx* in ASJ significantly inhibited the speed response to $H_2S$ (*Figure 2H*). Surprisingly, using the $Ca^{2+}$ sensor GCaMP6s to monitor a direct response of ASJ neurons to $H_2S$, we observed no $H_2S$-induced $Ca^{2+}$ transients under any of the tested conditions, whereas $CO_2$-evoked $Ca^{2+}$ increases were readily detected (*Figure 2I*, *Figure 2—figure supplement 2E*). These findings suggest that although ASJ neuronal activity and neurosecretion contribute to the $H_2S$ responses, ASJ neurons are unlikely to play a role in directly detecting $H_2S$.

## Starvation modulates $H_2S$ avoidance

In the candidate gene screen, another set of mutants with reduced speed responses to $H_2S$ had previously been shown to be defective in response to $CO_2$ (*Hallem and Sternberg, 2008*). Specifically, the $H_2S$-evoked speed response was reduced in mutants deficient in nutrient-sensitive signaling, including the insulin receptor DAF-2, the TGF-β ligand DAF-7, and the nuclear hormone receptor NHR-49 (*Figure 3A–C*, *Figure 3—figure supplement 1A*). Similarly, mutants lacking other $CO_2$ response modulators, such as the calcineurin subunits TAX-6 and CNB-1, also failed to respond to $H_2S$ (*Figure 3—figure supplement 1B, C*). In addition to the abolished speed response, we also observed that the $H_2S$-evoked omega-turn response was nearly abolished in *daf-2* mutants, and a high proportion of animals exhibited persistently high reversal rate regardless of $H_2S$ stimulation (*Figure 3—figure supplement 1D–F*). The $H_2S$ avoidance defects observed in *daf-2* and *daf-7* single mutants were fully suppressed in *daf-2; daf-16* and *daf-7; daf-3* double mutants (*Figure 3A and B*). Therefore, it is likely that insulin and TGF-β signaling modulate $H_2S$ responses by regulating the expression of relevant genes via DAF-16 and DAF-3 transcription factors, respectively. Given that $H_2S$-evoked speed response was modulated by the nutrient-sensitive pathways, we further explored whether the locomotory response to $H_2S$ was sensitive to the nutrient state. Similar to the response to $CO_2$ (*Bretscher et al., 2008*; *Hallem and Sternberg, 2008*), we observed that a 24-hr period of starvation suppressed the speed response to $H_2S$ (*Figure 3D*). However, the $CO_2$ sensor GCY-9 was dispensable for $H_2S$-evoked avoidance (*Figure 3—figure supplement 1G–I*). These findings suggest that while acute avoidance of $CO_2$ and $H_2S$ share common modulatory pathways, the responses are initiated through distinct mechanisms.

## The $O_2$ sensing circuit antagonizes $H_2S$ avoidance in wild isolates

Wild strains of *C. elegans* thrive in environmental niches characterized by variable $O_2$, $CO_2$, and $H_2S$ levels (*Adams et al., 1979*; *Bretscher et al., 2008*; *Budde and Roth, 2011*; *Gea et al., 2004*; *Hallem and Sternberg, 2008*; *Morra and Dick, 1991*; *Oshins et al., 2022*; *Patange et al., 2025*; *Rodriguez-Kabana et al., 1965*). Compared to the laboratory reference strain N2, wild isolates display robust responses to high $O_2$ but reduced sensitivity to $CO_2$ stimulation (*Beets et al., 2020*; *Bretscher et al., 2008*; *Carrillo et al., 2013*; *Hallem and Sternberg, 2008*; *Kodama-Namba et al., 2013*; *McGrath et al., 2009*). It is thought that the unique $O_2$ and $CO_2$ responses of N2 strain evolved during its domestication on solid media at atmospheric gas concentrations (*Sterken et al., 2015*). We hypothesized that wild *C. elegans* strains might also exhibit an attenuated response to $H_2S$, and therefore included a set of wild strains in our candidate gene screen. Consistent with this hypothesis, wild isolates including CB4855, CB4856, and CB4858 exhibited reduced locomotory speed in $H_2S$ (*Figure 3E*, *Figure 3—figure supplement 1J*).

A naturally occurring variation in the neuropeptide receptor NPR-1 predominantly determines the differences in locomotory responses to $O_2$ and $CO_2$ between the laboratory N2 strain and wild isolates. The NPR-1(215 F) variant found in wild isolates is less active than the NPR-1(215 V) found in N2 animals (*de Bono and Bargmann, 1998*). We wondered whether the variations in the *npr-1* gene also contribute to differences in the acute response to $H_2S$. The absence of *npr-1* was sufficient to reduce the speed response to $H_2S$, suggesting that active NPR-1 signaling promotes $H_2S$ avoidance (*Figure 3F*). NPR-1(215 V) primarily acts within the RMG interneurons to modulate responses to environmental stimuli by inhibiting signal output from these neurons (*Figure 3G*; *Laurent et al., 2015*; *Macosko et al., 2009*). Cell-specific expression of the highly active *npr-1(215 V)* variant in RMG neurons effectively restored the $H_2S$-evoked locomotory speed in *npr-1* null mutant worms (*Figure 3H*). Furthermore, optogenetic stimulation of RMG interneurons using channelrhodopsin-2

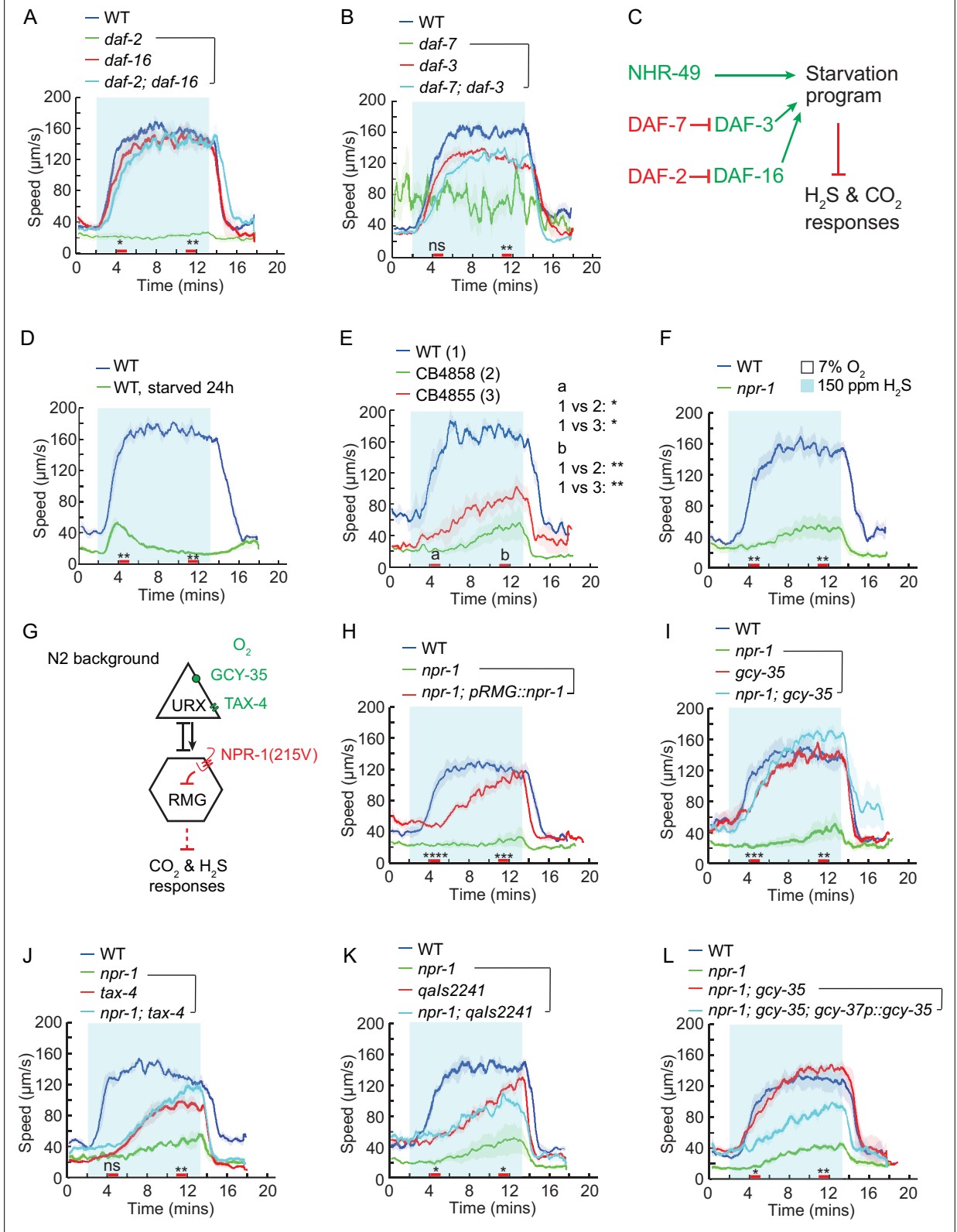

**Figure 3.** Insulin, TGF-β, and high $O_2$ signaling pathways antagonize $H_2S$ avoidance. (**A, B**) Locomotory speed responses to a switch from 7% $O_2$ to 150 ppm $H_2S$ balanced with 7% $O_2$ for animals of the indicated genotype: WT, *daf-2(e1370)*, *daf-16(mgDf47)*, and *daf-2(e1370); daf-16(mgDf47)* double mutants (**A**); WT, *daf-7(e1372)*, *daf-3(mgDf90)*, and *daf-7(e1372); daf-3(mgDf90)* double mutants (**B**). For the *daf-2* assays, WT and *daf-2* mutants were maintained at 15 °C. L4 animals were picked and shifted to 25 °C until day-1 adults, then assayed at room temperature. (**C**) Hypothetical model for

*Figure 3 continued on next page*

Figure 3 continued

the regulation of H$_2$S avoidance by a starvation program involving the insulin, TGF-β, and NHR-49 pathways. (D) Locomotory speed response to a switch from 7% O$_2$ to 150 ppm H$_2$S balanced with 7% O$_2$ in fed and starved WT animals. (E, F) Locomotory speed responses to a switch from 7% O$_2$ to 150 ppm H$_2$S balanced with 7% O$_2$ for animals of the indicated genotype: WT and wild isolates (CB4858 and CB4855) (E); WT and npr-1(ad609) (F). (G) Hypothetical model showing how O$_2$ regulates H$_2$S avoidance via the inhibitory RMG circuit activity, which is modulated by O$_2$ sensory inputs and NPR-1 signaling. (H–L) Locomotory speed responses to a switch from 7% O$_2$ to 150 ppm H$_2$S balanced with 7% O$_2$ for animals of the indicated genotype: WT, npr-1(ad609), and transgenic npr-1(ad609) expressing npr-1 in RMG neurons using Cre-LoxP system (*Macosko et al., 2009*) (H); WT, npr-1(ad609), gcy-35(ok769), and npr-1(ad609); gcy-35(ok769) double mutants (I); WT, npr-1(ad609), tax-4(p678) and npr-1(ad609); tax-4(p678) double mutants (J); WT, npr-1(ad609), qaIs2241(genetic ablation of AQR, PQR, and URX neurons) and npr-1(ad609); qaIs 2241 (K); WT, npr-1(ad609), npr-1(ad609); gcy-35(ok769) double mutants and transgenic npr-1(ad609); gcy-35(ok769) double mutants expressing gcy-35 cDNA under gcy-37 promoter, which drives gcy-35 expression in O$_2$ sensing neurons (L). For the comparison of acute locomotory responses between strains, red bars on the x-axis represent two intervals (4–5 min and 11–12 min, labeled a and b, respectively, in panel E) used for statistical analysis. ****=p < 0.0001, ***=p < 0.001, **=p < 0.01, *=p < 0.05, ns = not significant, Mann–Whitney U test.

The online version of this article includes the following figure supplement(s) for figure 3:

**Figure supplement 1.** Insulin, TGF-β, and O$_2$ signaling modulate locomotory response to H$_2$S.

(ChR2) significantly dampened the locomotory response to H$_2$S in N2 animals, suggesting that high RMG activity inhibits H$_2$S responses (*Figure 3—figure supplement 1K*).

The perception of 21% O$_2$ involves the soluble guanylate cyclases GCY-35/GCY-36 and the downstream cyclic nucleotide-gated (CNG) channels TAX-4/TAX-2 in the URX, AQR, and PQR O$_2$-sensing neurons, which relay the signals to RMG (*Busch et al., 2012*; *Cheung et al., 2005*; *Couto et al., 2013*; *Gray et al., 2004*; *Laurent et al., 2015*; *Persson et al., 2009*; *Zimmer et al., 2009*). We investigated whether disrupting the O$_2$ sensing machinery would have the same effect as inhibiting RMG signaling pathway by NPR-1(215 V). Mutating gcy-35 and tax-4, or genetically ablating the O$_2$ sensing neurons URX, AQR, and PQR by cell-specific expression of the pro-apoptotic gene egl-1 restored H$_2$S-evoked speed response in npr-1 null mutants (*Figure 3I–K*). Specific expression of gcy-35 cDNA in O$_2$ sensing neurons was sufficient to repress this response to H$_2$S in npr-1; gcy-35 double mutants (*Figure 3L*). Finally, enhancing the presynaptic activity of URX, AQR, and PQR neurons by the expression of a gain-of-function PKC-1(A160E) (*Dekker et al., 1993*; *Hiroki et al., 2022*) significantly dampened H$_2$S-induced locomotory speed (*Figure 3—figure supplement 1L*). Taken together, these observations indicate that activation of the O$_2$-sensing circuit inhibits the neural pathway required for H$_2$S-evoked avoidance response.

## H$_2$S exposure reprograms gene expression in *C. elegans*

Wild isolates that were relatively insensitive to H$_2$S exposure (*Figure 3E*, *Figure 3—figure supplement 1J*) likely evolved adaptations to persist in environments where H$_2$S levels may transiently increase (*Budde and Roth, 2011*; *Patange et al., 2025*; *Rodriguez-Kabana et al., 1965*; *Romanelli-Cedrez et al., 2024*). These adaptations to H$_2$S are mediated, at least in part, by the reprogramming of gene expression (*Miller et al., 2011*). To identify the genes whose expression was induced by prolonged H$_2$S exposure, we conducted a comparative analysis of transcriptome profiles in animals exposed to 50 ppm or 150 ppm H$_2$S for 1 hr, 2 hr, or 12 hr. RNA-seq analysis revealed that exposure to either 50 ppm or 150 ppm H$_2$S for one hour was sufficient to trigger significant changes in gene expression, with 518 or 304 genes showing differential expression, respectively (*Figure 4A*, *Supplementary file 2*). Genes induced by H$_2$S exposure for 1 or 2 hr displayed considerable overlap (*Figure 4B*, *Figure 4—figure supplement 1A*). Gene Ontology (GO) analysis revealed that biological processes such as defense against bacteria and cysteine biosynthesis from L-serine were significantly enriched after H$_2$S exposure (*Figure 4C and D*, *Figure 4—figure supplement 1B, C*). As expected, we observed a robust induction of genes involved in H$_2$S detoxification (*Figure 4E and F*; *Horsman and Miller, 2016*; *Miller et al., 2011*; *Niu et al., 2011*; *Vora et al., 2022*). H$_2$S is detoxified in mitochondria by sulfide:quinone oxidoreductase SQRD-1 to produce persulfide, which is then metabolized by sulfur dioxygenase ETHE-1 to generate sulfite. Thiosulfate transferase further oxidizes sulfite to produce thiosulfate (*Hildebrandt and Grieshaber, 2008*). Glutathione S-transferases (GSTs) remove accumulated sulfur molecules during H$_2$S oxidation (*Jackson et al., 2012*). Notably, gst-19 and sqrd-1 were among the most significantly upregulated genes after one or two hours of exposure to either 50 ppm

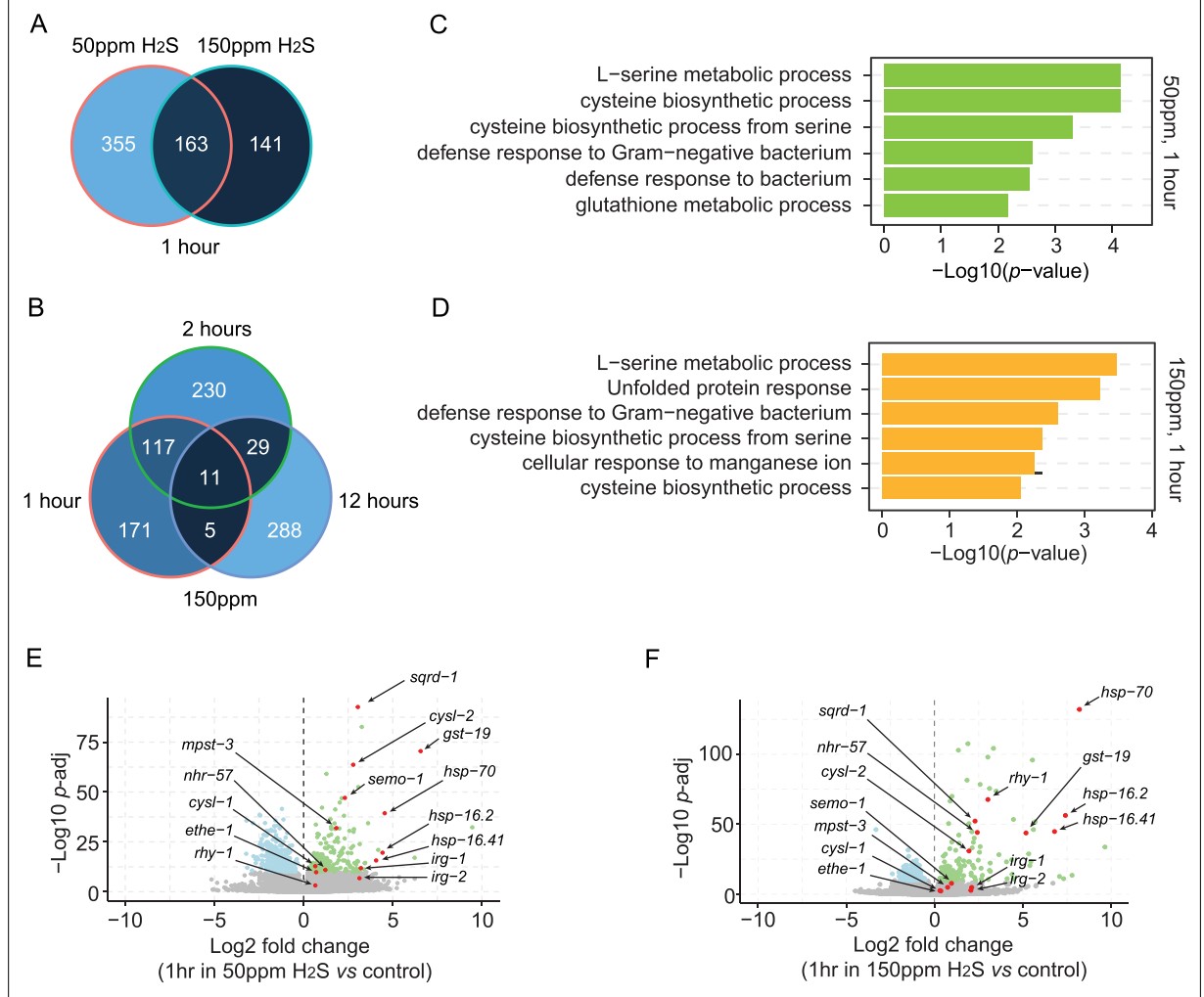

**Figure 4.** Prolonged H₂S exposure reprograms gene expression in *C. elegans*. (**A**) Venn diagram displaying the number of differentially expressed genes after 1 hour exposure to 50 ppm or 150 ppm H₂S balanced with 7% O₂ in WT animals (adjusted p<1e-10). (**B**) Venn diagram displaying the number of differentially expressed genes after 1-, 2-, and 12 hr exposure to 150 ppm H₂S balanced with 7% O₂ in WT animals (adjusted p<1e-10). (**C, D**) Significantly enriched GO categories for differentially expressed genes with adjusted p<1e-10 in WT animals exposed to 50 ppm (**C**) or 150 ppm (**D**) H₂S balanced with 7% O₂ for 1 hr. (**E, F**) Volcano plots showing the differentially expressed genes with adjusted p<1e-10 in WT animals exposed to 50 ppm (**E**) or 150 ppm (**F**) H₂S balanced with 7% O₂ for 1 hr. A set of genes involved in H₂S detoxification, cysteine metabolism, and stress response was highlighted in red.

The online version of this article includes the following figure supplement(s) for figure 4:

**Figure supplement 1.** Transcriptome reprogramming induced by H₂S exposure.

or 150 ppm H₂S, while *ethe-1* showed a weak increase (*Figure 4E and F*, *Figure 4—figure supplement 1D, E*, *Supplementary file 2*).

H₂S exposure is known to activate the HIF-1 pathway (*Budde and Roth, 2010*; *Budde and Roth, 2011*; *Horsman et al., 2019*; *Ma et al., 2012*; *Miller et al., 2011*; *Powell-Coffman, 2010*; *Topalidou and Miller, 2017*). In line with this, a proportion of genes with HIF-1 regulated promoters displayed increased expression after 1 or 2 hr of H₂S exposure (*Figure 4—figure supplement 1F*). Among these were the aforementioned detoxifying enzymes GST-19, SQRD-1, and ETHE-1, as well as CYSL-1, CYSL-2, and CDO-1, which are involved in cysteine metabolism (*Supplementary file 2*). Consistent with earlier studies, we also observed differential expression of a set of SKN-1 targets in response to chronic H₂S exposure (*Figure 4—figure supplement 1G*; *Horsman et al., 2019*; *Miller et al., 2011*; *Niu et al., 2011*). This included the heat shock protein genes *hsp-16.2* and *hsp-16.41* (*Figure 4E and F*, *Figure 4—figure supplement 1D, E*). Surprisingly, we detected a rapid increase in the expression of genes associated with intracellular H₂S production, including the methanethiol

oxidase-encoding gene *semo-1* and the mercaptopyruvate sulfurtransferase-encoding gene *mpst-3* (*Figure 4E and F*, *Figure 4—figure supplement 1D, E*; *Philipp et al., 2022*; *Qabazard et al., 2014*). Elevated expression of 3-mercaptopyruvate sulfurtransferase has similarly been reported in sulfide spring fish, suggesting a conserved response mechanism to $H_2S$ (*Kelley et al., 2016*; *Mathew et al., 2011*; *Qabazard et al., 2014*). However, the functional implications of increased expression of $H_2S$ synthesis genes remain unclear.

When exposed to $H_2S$ for 12 hr, the differentially expressed genes showed reduced overlap with those identified at earlier time points (*Figure 4B*, *Figure 4—figure supplement 1A*), suggesting that distinct defense strategies may be employed at various stages of $H_2S$ exposure. Furthermore, substantial overlap was observed between the sets of genes induced by 50 ppm and 150 ppm $H_2S$ after 1 hr or 2 hr exposure. However, this overlap decreased significantly after 12 hr exposure (*Figure 4—figure supplement 1H–J*). Additionally, we noticed that 150 ppm $H_2S$ specifically triggered the expression of the HIF-1 regulator *rhy-1* and the HIF-1 target *nhr-57* at all exposure time points, a response that was less pronounced at 50 ppm $H_2S$ (*Figure 4E and F*, *Figure 4—figure supplement 1D, E, K and L*). Together, these findings suggest that multiple defense mechanisms are induced to adapt to the stress imposed by $H_2S$, which can diverge over time and at different $H_2S$ concentrations.

## Locomotory speed response to $H_2S$ is modulated by HIF-1-induced detoxification

Given that HIF-1 signaling is robustly activated shortly after $H_2S$ exposure (*Figure 4—figure supplement 1F*; *Budde and Roth, 2010*; *Budde and Roth, 2011*; *Miller et al., 2011*), we next sought to determine how HIF-1 signaling might contribute to $H_2S$-evoked avoidance response and began by examining the effects of HIF-1 stabilization. Acclimating animals in 1% $O_2$ for 12 hr or longer significantly decreased the speed response to $H_2S$, indicating that HIF-1 stabilization inhibits the $H_2S$-evoked avoidance behavior (*Figure 5A*). To confirm this observation, we examined several mutants with stabilized HIF-1. The conserved proline-4-hydroxylase PHD/EGL-9 and the von Hippel-Lindau (VHL) tumor suppressor protein are required for HIF-1 degradation (*Kaelin and Ratcliffe, 2008*; *Semenza, 2010*). Disruption of either *egl-9* or *vhl-1* leads to HIF-1 stabilization under normoxic conditions. Similar to the effect of prolonged hypoxia, both mutants exhibited a markedly reduced locomotory speed in $H_2S$ (*Figure 5B*). Interestingly, the omega-turn and reversal responses to $H_2S$ were also substantially inhibited in *egl-9* mutants (*Figure 5—figure supplement 1A, B*). In contrast, the locomotory speed response to 1% $O_2$ remained largely unaffected in *egl-9* deficient animals (*Figure 5—figure supplement 1C*), suggesting a selective impairment of the $H_2S$ response. In addition, expressing the non-degradable forms of HIF-1, *hif-1*(P621A or P621G), under the pan-neuronal *unc-14* promoter or under the endogenous *hif-1* promoter, was sufficient to inhibit $H_2S$-evoked speed responses (*Figure 5C*). Together, these findings indicate that the activation of HIF-1 signaling during prolonged $H_2S$ or hypoxia exposure mediates an adaptive response associated with reduced behavioral responses to $H_2S$, and that HIF-1 stabilization in neurons alone is sufficient to promote this behavioral adaptation.

On the other hand, animals lacking HIF-1 signaling have previously been shown to display increased sensitivity to $H_2S$ and become paralyzed rapidly upon exposure (*Budde and Roth, 2010*; *Horsman et al., 2019*). Consistent with this, *hif-1* mutants exhibited a brief increase in speed upon $H_2S$ exposure, which rapidly declined to baseline levels (*Figure 5D*). They also displayed a stronger initial omega-turn response and greater inhibition of the reversal rate than WT (*Figure 5E and F*), confirming that animals deficient in HIF-1 signaling are sensitized to $H_2S$ exposure. This transient avoidance behavior was also observed in *egl-9; hif-1* and *vhl-1; hif-1* double mutants (*Figure 5G and H*), suggesting that this phenotype is linked to the absence of HIF-1 signaling. Importantly, *hif-1* mutants displayed a robust speed response to other stimuli, such as near-UV light and 1% $O_2$ stimulation (*Figure 5I*, *Figure 5—figure supplement 1D*), suggesting that the brief avoidance response is specific to $H_2S$ rather than reflecting a general locomotory defect. Further supporting increased sensitivity to $H_2S$ in *hif-1* mutants, the locomotory speed response of *hif-1* mutants to subsequent near-UV light stimulation was nearly abolished after 30 min of $H_2S$ exposure, whereas *egl-9* mutants and WT animals remained fully responsive (*Figure 5I*). The reduced movement and lack of responsiveness to other stressors in *hif-1* mutants resemble the stress-induced sleep-like behavior previously observed in response to noxious heat (*Byrne Rodgers and Ryu, 2020*). Prompted by this observation, we next explored whether HIF-1-induced detoxification genes were also required to maintain $H_2S$-evoked

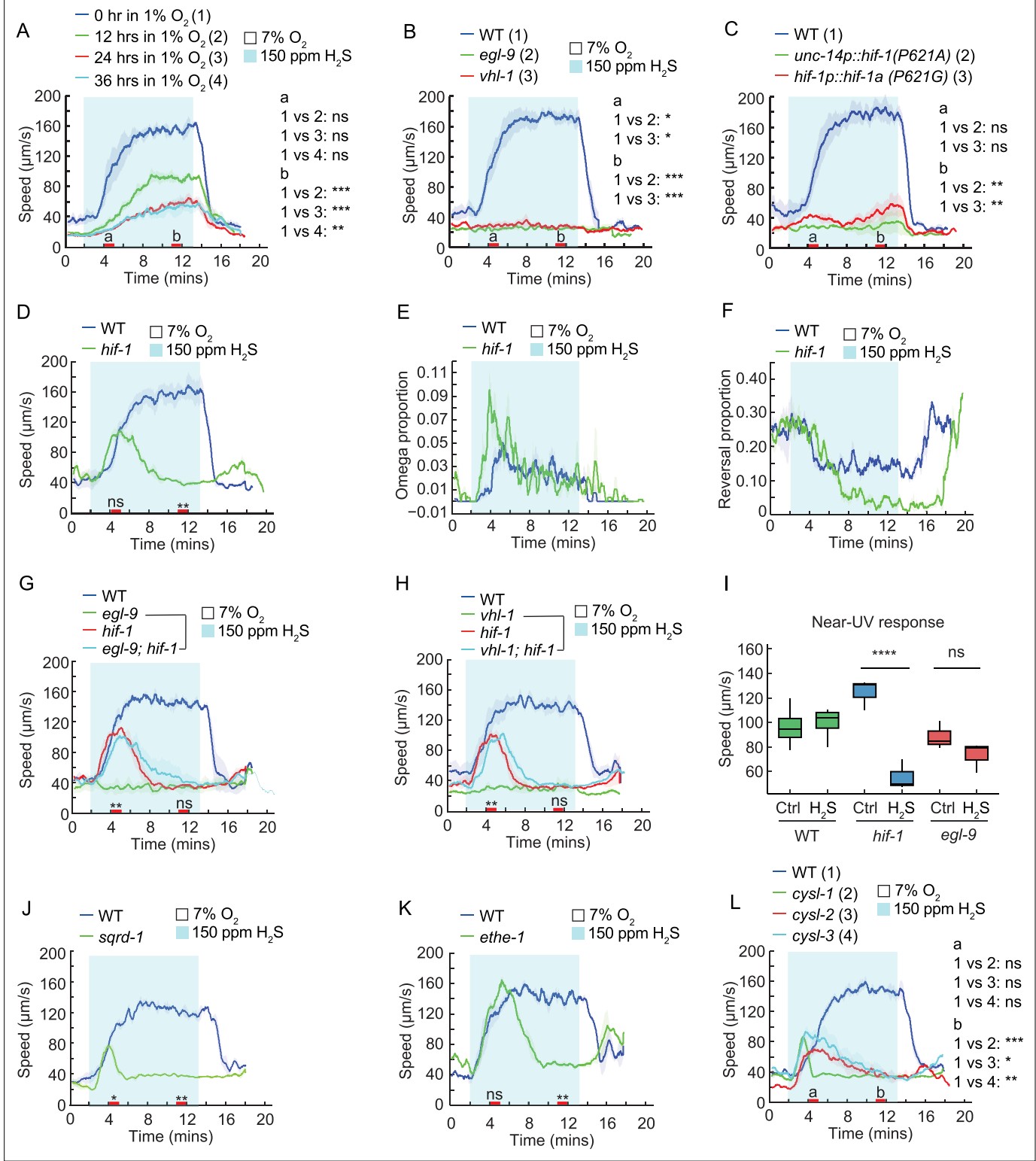

**Figure 5.** Acute response to H₂S is modulated by HIF-1 signaling. (**A**) Locomotory speed responses of WT animals to a switch from 7% O₂ to 150 ppm H₂S balanced with 7% O₂ after incubation of the animals for 0, 12, 24, or 36 hr in 1% O₂. (**B, C**) Locomotory speed responses to a switch from 7% O₂ to 150 ppm H₂S balanced with 7% O₂ for animals of the indicated genotype: WT, *egl-9(sa307)*, and *vhl-1(ok161)* (**B**); WT and transgenic animals expressing the non-degradable form of HIF-1 (P621A or P621G) under the pan-neuronal *unc-14* promoter or the endogenous *hif-1* promoter, respectively (**C**). (**D–F**) Avoidance responses of WT and *hif-1(ia4)* to a switch from 7% O₂ to 150 ppm H₂S balanced with 7% O₂, including locomotory speed (**D**), reorientation (omega-turn) (**E**), and reversal (**F**). (**G, H**) Locomotory speed responses to a switch from 7% O₂ to 150 ppm H₂S balanced with 7% O₂ for animals of the

*Figure 5 continued on next page*

Figure 5 continued

indicated genotype: WT, *egl-9(sa307)*, *hif-1(ia4)*, and *egl-9(sa307); hif-1(ia4)* double mutants (**G**); WT, *vhl-1(ok161)*, *hif-1(ia4)*, and *vhl-1(ok161); hif-1(ia4)* double mutants (**H**). (**I**) Average locomotory speed during 2 min of exposure to near-UV light for WT, *hif-1(ia4)*, and *egl-9(sa307)* before (Ctrl = control) and after 30 min of exposure to 150 ppm $H_2S$ balanced with 7% $O_2$. (**J–L**) Locomotory speed responses to a switch from 7% $O_2$ to 150 ppm $H_2S$ balanced with 7% $O_2$ for animals of the indicated genotype: WT and *sqrd-1(tm3378)* (**J**); WT and *ethe-1(yum2895)* (**K**); WT, *cysl-1(ok762), cysl-2(ok3516)*, and *cysl-3(yum4)* (**L**). For the comparison of acute locomotory responses between strains, red bars on the x-axis represent two intervals (4–5 minutes and 11–12 minutes, labeled a and b, respectively, in panels A, B, C, and L) used for statistical analysis. \*\*\*=p < 0.001, \*\*=p < 0.01, \*=p < 0.05, ns = not significant, Mann–Whitney U test.

The online version of this article includes the following figure supplement(s) for figure 5:

**Figure supplement 1.** Acute response to $H_2S$ is modulated by HIF-1 signaling.

behavioral responses (*Figure 5—figure supplement 1E*). Similar to *hif-1* mutants, *sqrd-1* and *ethe-1* mutants displayed brief responses to $H_2S$, with locomotory activity rapidly returning to baseline levels (*Figure 5J and K*, *Figure 5—figure supplement 1F–I*). However, both mutants responded normally to 1% $O_2$, suggesting a specific hypersensitivity to $H_2S$ toxicity (*Figure 5—figure supplement 1J*). Furthermore, we examined the contribution of additional genes whose expression was upregulated in response to $H_2S$, including the sulfhydrylase/cysteine synthase-encoding genes *cysl-1*, *cysl-2*, and *cysl-3*, as well as *semo-1*. Disrupting any of these genes phenocopied the responses observed in *hif-1*, *ethe-1*, or *sqrd-1* mutants, while they displayed normal responses to 1% $O_2$ (*Figure 5L*, *Figure 5—figure supplement 1D, K–P*). These observations suggest that while the HIF-1-induced $H_2S$ detoxification system is not essential for initiating the response to $H_2S$, it is required to maintain animals' locomotory activity in the presence of $H_2S$.

## Labile iron pool sustains the locomotory activity in $H_2S$

Among the most differentially regulated genes upon exposure to $H_2S$, we observed consistent down-regulation of *ftn-1* and, to a lesser extent, *ftn-2*, whereas *smf-3* expression was increased under specific $H_2S$ conditions (*Figure 6A and B*, *Figure 6—figure supplement 1A–D*). The ferritin-encoding genes *ftn-1* and *ftn-2*, along with the iron transporter *smf-3*, are critical for maintaining intracellular labile iron homeostasis in *C. elegans* (*Ackerman and Gems, 2012*; *Gourley et al., 2003*; *Rajan et al., 2019*; *Romney et al., 2011*), suggesting that $H_2S$ exposure may interfere with cellular iron storage. It has been shown that disrupting *ftn-1* increases labile iron pools in the cytoplasm, while its overexpression reduces these levels (*Anderson and Leibold, 2014*; *Romney et al., 2011*). We found that *ftn-1* mutants had enhanced speed response to $H_2S$, whereas *ftn-1* overexpression attenuated $H_2S$-induced speed increase without damaging acute response to 1% $O_2$ (*Figure 6C and D*, *Figure 6—figure supplement 1E*). In addition, *smf-3* mutants, which impair iron uptake, exhibited a reduced speed response to $H_2S$ (*Figure 6D*). These results support the idea that $H_2S$ exposure disrupts iron homeostasis.

To complement our genetic analysis, we pharmacologically depleted labile irons. In the presence of the iron chelator 2,2'-dipyridyl (BP), the $H_2S$-evoked speed response was fully lost (*Figure 6E*), whereas locomotory speed in response to hypoxia was maintained (*Figure 6—figure supplement 1F*). We noticed that, similar to *hif-1* mutants, both *smf-3* mutants and BP-treated animals exhibited a strong initial omega-turn response to $H_2S$ (*Figure 6—figure supplement 1G, I*), suggesting that these animals are hypersensitive to $H_2S$ toxicity. One possibility is that low iron levels compromise $H_2S$ detoxification capacity, as the detoxification enzyme ETHE-1 requires iron for its activity (*Kabil and Banerjee, 2012*; *Pettinati et al., 2015*). By contrast, increasing iron levels by feeding ferric ammonium citrate (FAC) or by disrupting *ftn-1* significantly sustained animals' locomotory speed in $H_2S$ (*Figure 6D and E*), likely by delaying $H_2S$-mediated iron depletion. However, the reorientation and reversals were not significantly affected (*Figure 6—figure supplement 1G–J*). This impact of FAC supplementation and *ftn-1* disruption on speed response was largely lost in the absence of *hif-1* (*Figure 6F and G*), suggesting that the effects of iron require HIF-1-dependent detoxification (*Figure 6H*). Yet, increased iron availability modestly improved the locomotory activity of *hif-1* mutants in $H_2S$ (*Figure 6F and G*), presumably by compensating for the low iron levels in *hif-1* mutant animals (*Rajan et al., 2019*; *Romney et al., 2011*).

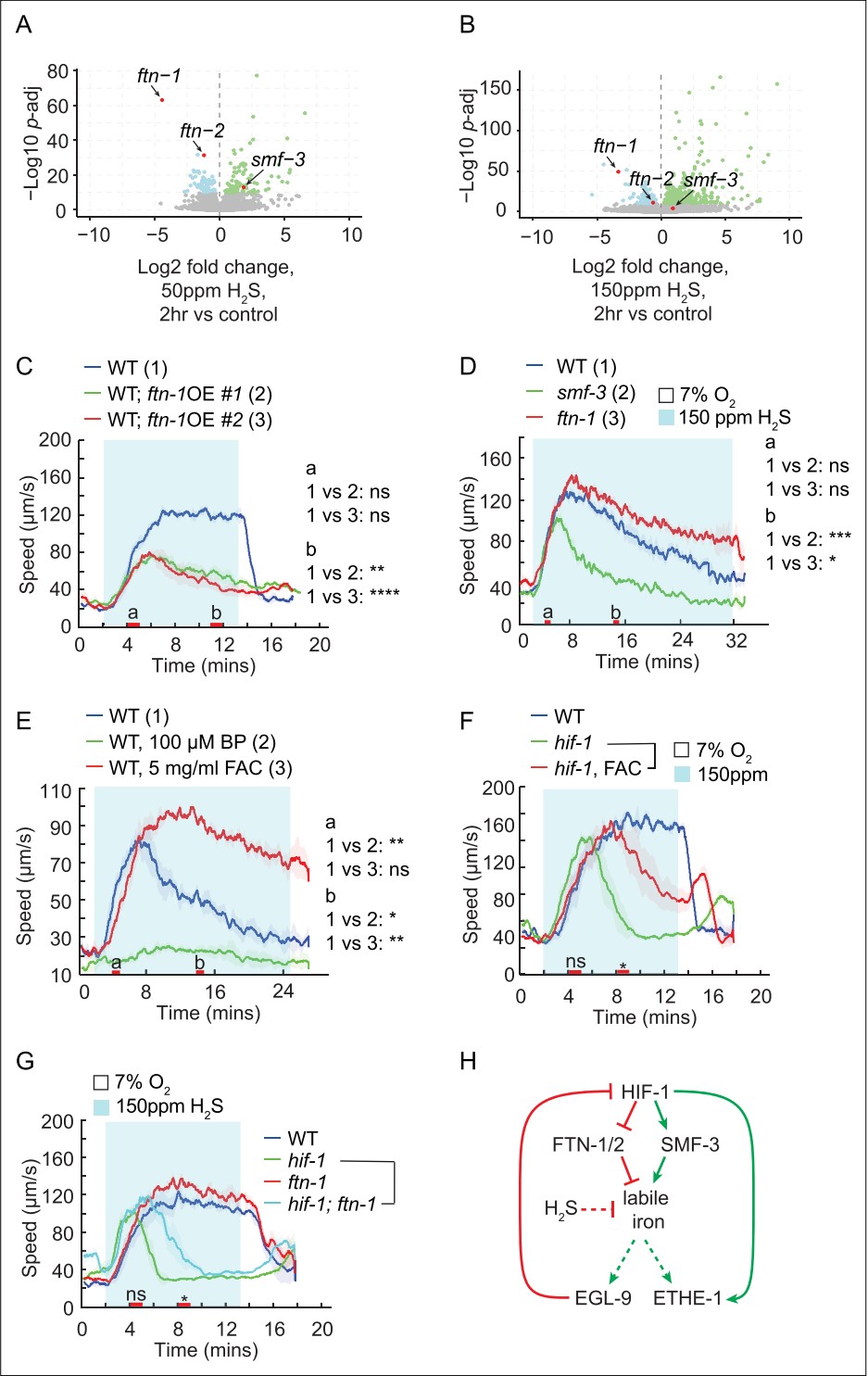

**Figure 6.** Labile iron pool sustains the locomotory activity in H$_2$S. (**A, B**) Volcano plots showing the relative expression of the genes involved in the regulation of iron homeostasis in WT animals after 2 hr exposure in 50 ppm (**A**) or 150 ppm (**B**) H$_2$S balanced with 7% O$_2$. (**C–G**) Locomotory speed responses to a switch from 7% O$_2$ to 150 ppm H$_2$S balanced with 7% O$_2$ for animals of indicated genotypes or treatments: WT and animals overexpressing *ftn-1* genomic DNA under its own promoter (#1 and #2 indicate two independent lines) (**C**); WT, *smf-3(ok1305)*, and *ftn-1(ok3625)* (**D**); WT, and WT pretreated with 100 µM 2,2'-Bipyridyl (BP) or with 5 mg/ml ferric ammonium citrate (FAC) in the presence of food for 16 hr (**E**); WT, *hif-1(ia4)*, and *hif-1(ia4)* mutants pretreated with 5 mg/ml ferric ammonium citrate (FAC) for 16 hr (**F**); WT, *hif-1(ia4)*, *ftn-1(ok3625)*, and *hif-1(ia4); ftn-1(ok3625)* double mutants

*Figure 6 continued on next page*

*Figure 6 continued*

(**G**). (**H**) Hypothetical model of the interactions between labile iron pool and HIF-1 signaling. For the comparison of acute locomotory speed responses between strains or conditions, red bars on the x-axis represent two intervals (4–5 min and 11–12 min, labeled a and b, respectively, in panels C, D, and E) used for statistical analysis. ****=p < 0.0001, ***=p < 0.001, **=p < 0.01, *=p < 0.05, ns = not significant, Mann–Whitney U test.

The online version of this article includes the following figure supplement(s) for figure 6:

**Figure supplement 1.** Labile iron pool sustains the locomotory activity in $H_2S$.

## Mitochondrial function modulates $H_2S$-evoked locomotion responses

The mitochondrial electron transport chain (ETC) is not only the primary target of $H_2S$ toxicity, but is also an essential component of the $H_2S$ detoxification pathways (*Cooper and Brown, 2008*; *Khan et al., 1990*; *Nicholls and Kim, 1982*; *Romanelli-Cedrez et al., 2024*). Based on this, we hypothesized that ETC plays a central role in the locomotory response to $H_2S$. Indeed, mutating the genes critical for ETC function, including *gas-1*, *clk-1*, *mev-1*, and *isp-1*, impaired the $H_2S$-evoked speed and reversal responses (*Figure 7A–D, Figure 7—figure supplement 1A–D*). These data suggest either that dysfunctional ETC induces an adaptive response to cellular stressors or that a functional ETC is required to trigger and support $H_2S$-evoked avoidance behavior. Consistent with the later model, exposing animals to rotenone, an ETC complex I inhibitor, robustly inhibited the speed responses to $H_2S$ at all time points we tested (*Figure 7E*). Similar to ETC mutants, WT animals exposed to rotenone for 2 hr also showed reduced basal locomotory activity (*Figure 7E*).

In contrast to prolonged rotenone exposure, we noticed that transient rotenone exposure substantially increased animals' basal locomotory speed (*Figure 7E*; *Onukwufor et al., 2022*), suggesting that acute ETC interruption evokes behavioral avoidance, in a manner similar to that observed with other noxious stimuli. Since rotenone rapidly and persistently stimulates mitochondrial ROS production (*Ochi et al., 2016*; *Ramsay and Singer, 1992*; *Zorov et al., 2014*), the transient locomotory increase during short rotenone exposure (*Figure 7E*) is likely driven by these ROS bursts, whereas reduced locomotory activity during prolonged exposure is presumably caused by chronically elevated ROS production. As toxic levels of $H_2S$ also induce ROS (*Jia et al., 2020*), this raised the possibility that $H_2S$-evoked behavioral responses were modulated by the mitochondrial ROS. Therefore, we sought to further explore how mitochondrial ROS contributes to speed response to high $H_2S$. We focused on superoxide dismutases (SODs), key enzymes in superoxide detoxification. The *C. elegans* genome encodes five *sod* genes: *sod-1* and *sod-5* encode cytosolic Cu/ZnSODs, *sod-2* and *sod-3* encode mitochondrial MnSODs, and *sod-4* encodes the extracellular Cu/ZnSOD isoforms (*Doonan et al., 2008*; *Zubovych et al., 2010*). All *sod* single mutants exhibited a robust initial response to 150 ppm $H_2S$ (*Figure 7F and G*). Interestingly, the mitochondrial SOD mutants *sod-2* and *sod-3*, as well as *sod-5*, showed an initial burst of locomotory activity followed by a rapid decline in speed in the presence of $H_2S$ (*Figure 7F and G*), but maintained a high locomotory speed in response to 1% $O_2$ (*Figure 7—figure supplement 1E*). In the quintuple *sod-1; sod-2; sod-3; sod-4; sod-5* mutant, which lacks all superoxide dismutases and presumably accumulates higher ROS levels, the $H_2S$-evoked speed and omega-turn responses were nearly abolished, and the proportion of animals exhibiting reversals remained consistently high, a pattern resembling that of *egl-9* and ETC mutants (*Figure 7G, Figure 7—figure supplement 1F, G*). Similar to *sod-2* and *sod-3*, the quintuple *sod* mutant exhibited robust speed response to 1% $O_2$ (*Zhao et al., 2022*). In addition, the quintuple mutants responded normally to near-UV light after 30 min of $H_2S$ exposure (*Figure 7—figure supplement 1H*), indicating that their neuromuscular function remains largely preserved. These observations suggest that either constitutively high ROS levels in quintuple mutants dampen $H_2S$-evoked ROS transients, or they promote adaptation to $H_2S$-induced stress, presumably via activation of stress-responsive pathways such as HIF-1, NRF2/SKN-1, and DAF-16/FOXO, ultimately reducing the locomotory responses (*Figure 7H*; *Lee et al., 2010*; *Lennicke and Cochemé, 2021*; *Patten et al., 2010*).

## Discussion

Animals' behavior and physiology are profoundly influenced by the environments in which they evolved. The laboratory strain N2, which is adapted to low atmospheric $CO_2$ and $H_2S$ concentrations, robustly avoids both gases (*Beets et al., 2020*; *Bretscher et al., 2008*; *Carrillo et al., 2013*; *Hallem*

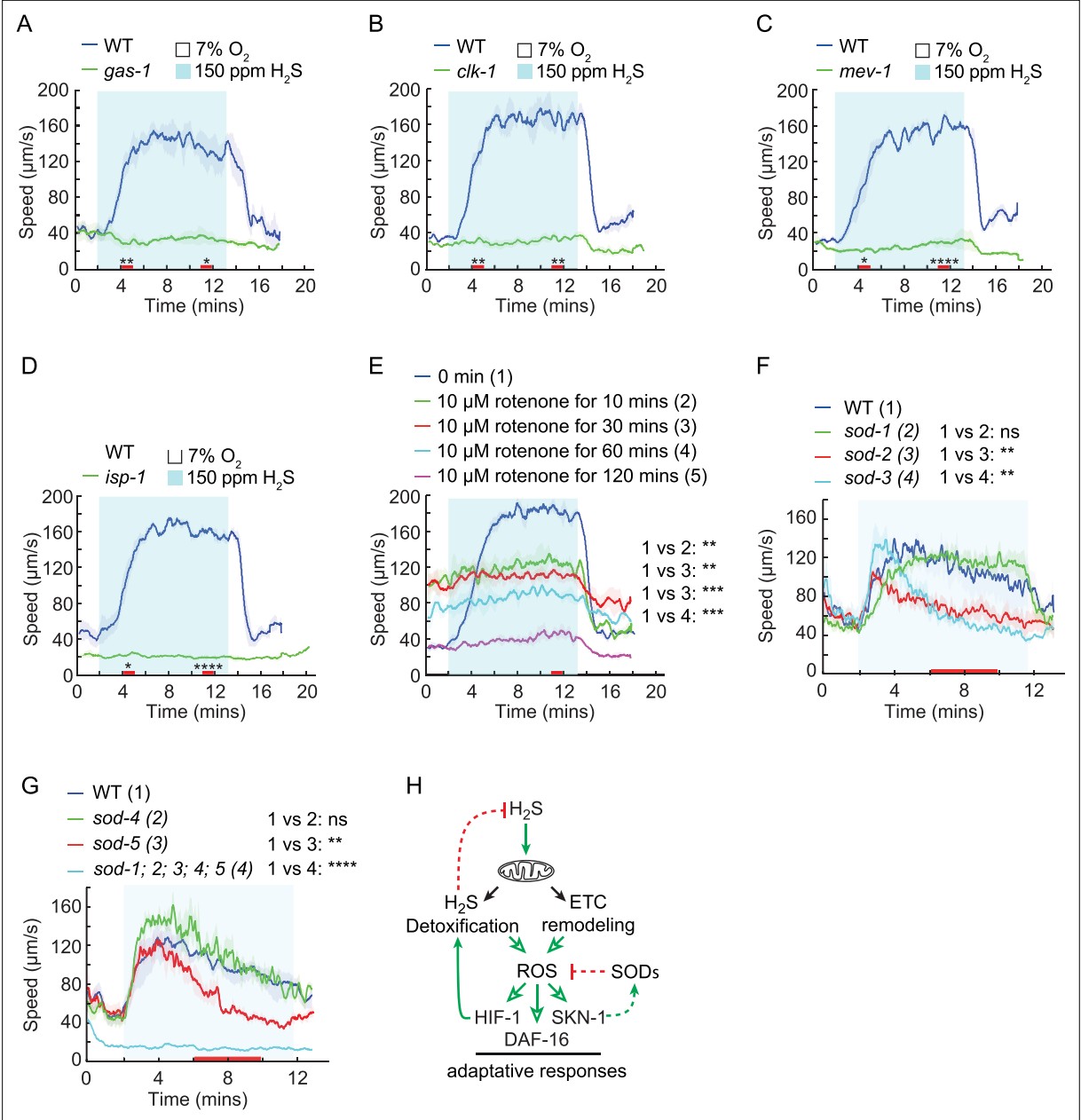

**Figure 7.** Mitochondrial function is required for acute response to H₂S. (**A–D**) Locomotory speed responses to a switch from 7% O₂ to 150 ppm H₂S balanced with 7% O₂ for animals of the indicated genotype: WT and *gas-1(fc21)* (**A**); WT and *clk-1(qm30)* (**B**); WT and *mev-1(kn1)* (**C**); WT and *isp-1(qm150)* (**D**). (**E**) Locomotory speed responses to a switch from 7% O₂ to 150 ppm H₂S balanced with 7% O₂ for WT animals pretreated with 10 μM rotenone for 0 min, 10 min, 30 min, 60 min, or 120 min. For the comparison of acute locomotory speed responses between strains or conditions, red bars on the x-axis represent two intervals (4–5 min and 11–12 min) used for statistical analysis. \*\*\*=p < 0.001, \*\*=p < 0.01, \*=p < 0.05, ns = not significant, Mann–Whitney U test. (**F–G**) Locomotory speed responses to a switch from 7% O₂ to 150 ppm H₂S balanced with 7% O₂ for animals of the indicated genotype: WT, *sod-1(tm776)*, *sod-2(ok1030)*, and *sod-3(tm760)* (**F**); WT, *sod-4(gk101)*, *sod-5(tm1146)*, and *sod-1(tm783); sod-2(ok1030); sod-3(tm760); sod-4(gk101); sod-5 (tm1146)* (**G**). For the comparison of acute locomotory speed responses between strains or conditions, red bars on the x-axis represent the time interval (6–10 min) used for statistical analysis. \*\*\*=p < 0.001, \*\*=p < 0.01, \*=p < 0.05, ns = not significant, Mann–Whitney U test. (**H**) Hypothetical model of the role of mitochondria in response to toxic levels of H₂S. In this model, mitochondria play a dual role in H₂S-evoked avoidance behavior. A transient burst of mitochondrial ROS triggered by high H₂S levels initiates locomotory avoidance, whereas sustained ROS elevation activates stress-responsive pathways, including HIF-1, NRF2/SKN-1, and DAF-16/FOXO, promoting adaptation to prolonged H₂S exposure.

The online version of this article includes the following figure supplement(s) for figure 7:

**Figure supplement 1.** Mitochondrial function is required for acute response to H₂S.

*and Sternberg, 2008*; *Kodama-Namba et al., 2013*; *McGrath et al., 2009*). Upon acute exposure to $H_2S$ above 75 ppm, N2 exhibits an avoidance behavior characterized by reorientation and increased locomotion. This response is significantly reduced in wild isolates. These divergent responses between the laboratory strain and wild isolates are primarily driven by variations in the *npr-1* gene, which encodes a neuropeptide receptor, suggesting that *C. elegans* has undergone rapid evolutionary adaptation to its environment.

The behavioral response to a specific sensory stimulus is usually shaped by an interplay of multiple environmental and physiological cues. We observe that conditions dampening *C. elegans'* response to $CO_2$ also impair the response to $H_2S$. Specifically, the speed response to $H_2S$ is significantly inhibited at high $O_2$ levels, and in mutants with defective insulin or TGF-β pathway that signal starvation. However, despite these shared modulations, $CO_2$ and $H_2S$ responses involve distinct molecular mechanisms. For example, $CO_2$ responses are mediated by the guanylate cyclase GCY-9 in BAG neurons (*Hallem et al., 2011*), which is dispensable for $H_2S$ responses. In addition, acute exposure to $H_2S$ induces a delayed acceleration compared to $CO_2$. These data suggest that while responses to the two gases share a downstream neural circuit that can be dynamically modulated by the state of $O_2$ sensing circuit, they are triggered by distinct mechanisms.

In a candidate gene survey, we excluded the potential involvement of globins, $K^+$ channels, biogenic amines, and most guanylate cyclases in $H_2S$ responses. However, we observed that $H_2S$ avoidance requires the activity of guanylate cyclase DAF-11 in ASJ neurons, as well as neurosecretion from these neurons. Since the DAF-11 pathway and neurosecretion from ASJ neurons regulate developmental programs that modify sensory functions in *C. elegans* (*Murakami et al., 2001*), it is not surprising that *daf-11* mutants display pleiotropic phenotypes including impaired $H_2S$ and $CO_2$ responses (*Hallem and Sternberg, 2008*). In addition, we did not detect an $H_2S$-evoked calcium transient in ASJ neurons. Therefore, although ASJ neurons and DAF-11 activity are clearly required, how $H_2S$ avoidance is triggered remains to be elucidated.

Consistent with previous reports (*Horsman et al., 2019*; *Miller et al., 2011*), $H_2S$ exposure substantially alters the gene expression in *C. elegans*, including those involved in $H_2S$ detoxification and iron homeostasis pathways. The repression of *ftn-1* and induction of *smf-3* observed during $H_2S$ exposure suggests that $H_2S$ depletes intracellular labile iron. Iron is a critical cofactor required for the activity of many enzymes and supports a wide range of cellular processes (*Dev and Babitt, 2017*; *Hentze et al., 2004*). One important enzyme that requires labile iron as a cofactor is PHD/EGL-9, which targets HIF-1 for degradation. Changes in iron availability are therefore expected to modulate EGL-9 activity, which in turn alters the expression of HIF-1-regulated genes (*Liochev, 1996*; *Myllyharju and Kivirikko, 1997*; *Read et al., 2021*; *Xu and Møller, 2011*). However, ETHE-1, a key $H_2S$ detoxification enzyme upregulated upon HIF-1 stabilization, also requires iron for its persulfide dioxygenase activity (*Kabil and Banerjee, 2012*; *Pettinati et al., 2015*). Therefore, $H_2S$-evoked responses under varying iron availability are determined by the combined effects of labile iron on HIF-1 signaling and on the $H_2S$ detoxification enzymes, among other factors. For instance, although reduced iron availability in *smf-3* or BP-treated animals may promote the HIF-1 induced detoxification pathways by inactivating EGL-9, the actual enzymatic clearance of $H_2S$ is impaired due to low ETHE-1 activity. Supporting this, *smf-3* mutants and BP-treated animals display increased sensitivity to $H_2S$, including enhanced initial omega-turn responses and rapid inhibition of locomotion. In contrast, increased iron availability in *ftn-1* mutants or FAC-treated animals likely delays iron depletion during $H_2S$ exposure, preserving ETHE-1 detoxification capacity and postponing the onset of HIF-1-mediated adaptations that are associated with reduced locomotion. This may explain the sustained high locomotory speed observed in *ftn-1* mutants or FAC-treated animals. In the absence of HIF-1, iron supplementation only partially improves the locomotory response to $H_2S$, likely because the $H_2S$ detoxification system including ETHE-1 cannot be transcriptionally induced. The modest effect observed may instead reflect correction of iron deficiency in *hif-1* mutants.

At concentrations below 50 ppm, $H_2S$ is well tolerated or even preferred, with beneficial effects such as improved thermotolerance (*Fawcett et al., 2015*; *Miller and Roth, 2007*; *Qabazard et al., 2014*). Notably, behavioral avoidance is absent below 50 ppm $H_2S$, suggesting that escape behavior is triggered only when $H_2S$ is not efficiently detoxified. This led us to hypothesize that animals with enhanced detoxification capacity would show reduced avoidance of otherwise toxic $H_2S$ levels. Indeed, the speed response to $H_2S$ is attenuated when the detoxification program is

upregulated under conditions of constitutive activation of HIF-1, such as in *egl-9* or *vhl-1* mutants, after prolonged exposure to 1% $O_2$, or when HIF-1 signaling is specifically activated in neurons. These observations support the idea that the initial avoidance response to $H_2S$ is triggered by neuronal detection of acute toxicity, while the subsequent decline in speed reflects a rapid activation of adaptive mechanisms, particularly through stabilization of HIF-1. The fact that wild-type animals remain responsive to other stimuli after prolonged $H_2S$ exposure suggests that reduced locomotory speed in $H_2S$ reflects active desensitization rather than general paralysis. However, persistently high $H_2S$ exposure is likely to exhaust cellular defense systems, leading to toxicity and paralysis. By contrast, the rapid loss of locomotion observed in *hif-1* mutants and detoxification-defective mutants is mediated by a mechanism distinct from adaptation. Their loss of responsiveness to other stimuli after $H_2S$ exposure suggests that these animals likely become sensitized and rapidly intoxicated by $H_2S$ due to impaired detoxification. Therefore, $H_2S$ exposure promotes a behavioral program that includes an initial reorientation and acceleration responses followed by a progressive adaptation driven by cellular detoxification processes. A similar behavioral program has been observed in *C. elegans* during noxious heat exposure, which induces short-term heat avoidance followed by long-lasting cytoprotective adaptation and a gradual reduction in avoidance (*Byrne Rodgers and Ryu, 2020*).

The interaction of $H_2S$ with mitochondrial ETC is multifaceted, acting as an electron donor at low concentrations and becoming a potent toxicant at high levels (*Szabo et al., 2014*), in part by promoting superoxide generation through complex IV inhibition and reverse electron transport at complex I (*Cooper and Brown, 2008*; *Jia et al., 2020*; *Khan et al., 1990*; *Nicholls and Kim, 1982*; *Romanelli-Cedrez et al., 2024*). We propose that mitochondria play a dual role in $H_2S$-evoked locomotory avoidance. On one hand, the mitochondrial ETC contributes to $H_2S$ detoxification and promotes adaptation. On the other hand, toxic levels of $H_2S$ remodel the ETC, leading to increased ROS production, which may serve as a trigger for the avoidance response. Supporting the idea that acute mitochondrial ROS generation initiates avoidance of high $H_2S$ levels, short-term rotenone exposure, known to promote mitochondrial ROS formation (*Ochi et al., 2016*; *Ramsay and Singer, 1992*; *Zorov et al., 2014*), substantially increases locomotory speed (*Onukwufor et al., 2022*). Meanwhile, the speed response to high $H_2S$ is fully suppressed by rotenone. This inhibition could result either from excessive mitochondrial ROS generated by rotenone, which may dampen the $H_2S$-triggered ROS spike, or from direct complex I inhibition, which may disrupt other $H_2S$ signaling processes required to initiate avoidance. However, persistent mitochondrial ROS production appears to suppress high locomotory speed and inhibit responsiveness to $H_2S$, as observed after 2 hr rotenone exposure, in mitochondrial ETC mutants, and in animals lacking all superoxide dismutases. One likely explanation is that mitochondrial ROS can activate a variety of stress-responsive pathways, including HIF-1, NRF2/SKN-1, and DAF-16/FOXO signaling (*Lee et al., 2010*; *Lennicke and Cochemé, 2021*; *Patten et al., 2010*), priming animals' adaptation to prolonged stress rather than causing toxicity. This is supported by the observation that even though SOD-deficient animals do not display strong initial locomotory responses to $H_2S$, they remain responsive to other stimuli after 30 min of $H_2S$ exposure, suggesting that high ROS levels do not compromise general viability or the $H_2S$ detoxification capacity. Therefore, we favor a model in which mitochondrial ROS exert a biphasic effect on $H_2S$-induced avoidance, facilitating $H_2S$ avoidance under acute conditions and contributing to locomotory inhibition when it is chronically elevated. Overall, lack of a clear sensory machinery, the slow increase of locomotory speed in $H_2S$ (*Figure 1D*), the rotenone-evoked speed responses, and the strong modulation of $H_2S$ responses by mitochondrial ETC inhibition suggest that $H_2S$ may not be directly perceived by *C. elegans*. Instead, acute avoidance to $H_2S$ is likely initiated by ROS-induced toxicity.

In summary, this study unveils a novel behavior of *C. elegans* in their avoidance to $H_2S$. The speed response to $H_2S$ is intricately shaped by environmental context, integrating inputs from other sensory cues such as food availability and $O_2$ levels. Our findings suggest that $H_2S$-induced locomotion arises from acute remodeling of mitochondrial ETC and associated ROS production, while highlighting the vital role of the HIF-1-induced detoxification pathways and iron homeostasis in protecting against $H_2S$-induced mitochondrial toxicity. Further work is needed to validate this model and to elucidate the neural circuits mediating the behavioral response to ROS-induced toxicity.

## Materials and methods

### Strains

The model organism *C. elegans* was used in this study. The N2 wild-type strain and mutant worms were obtained from the *Caenorhabditis* Genetics Center and the National BioResources Project Japan and maintained using standard protocols (*Brenner, 1974*). Strains used in this study are listed in *Supplementary file 1* and *Supplementary file 3*.

### Preparation of H₂S gas mixture

Different $H_2S$ concentrations were created as previously described (*Miller and Roth, 2007*). Briefly, the $H_2S$-containing gas mixture was prepared by diluting 5000 ppm $H_2S$ in nitrogen ($N_2$) with 7% $O_2$ balanced with $N_2$. The gas flow was tuned by Sierra Smart-Trak 100 mass flow controllers. $H_2S$ concentration in the mixture was measured using two $H_2S$ detectors (MSA ALTAIR 2 X gas detector for $H_2S$ and Clip Single Gas Detector, SDG, CROWCON). The pre-defined gas mixtures of 7% $O_2$, 1% $O_2$, and 5% $CO_2$ with $N_2$ were purchased from Air Liquide Gas AB. 5000 ppm $H_2S$ stock in $N_2$ was obtained from Linde Gas AB. Gas mixtures in all experiments were hydrated before use.

### Molecular biology

The Multisite Gateway system (Thermo Fisher Scientific, United States) was used to generate expression vectors. Promoters, including *gcy-37* (2.7 kb), *trx-1* (1 kb), *ftn-1*(2 kb), *daf-11*(3 kb), *odr-3*(2.7 kb), *gpa-11*(3 kb), *sra-6*(3 kb), *odr-1*(2.4 kb), *flp-21*(4.1 kb), and *ocr-2*(2.4 kb) were amplified from N2 genomic DNA and cloned into pDONR P4P1 (Thermo Fisher Scientific) using BP clonase. g*cy-35*, *tax-4,* and *pkc-1* cDNAs were amplified using the first strand cDNA library as the template, while *daf-11* and *ftn-1* genomic sequences were amplified using genomic DNA as the template and cloned into pDONR 221 (Thermo Fisher Scientific, 12536017) using BP reaction. To generate the gain-of-function mutation of *pkc-1* (A160E), Q5 Mutagenesis Kit (NEB) was used according to manufacturer instructions. All expression vectors were generated using LR reaction. The primer sequences were displayed in *Supplementary file 4*.

To generate transgenic animals, the *daf-11* expression vectors were injected at the concentration of 20 ng/μl supplemented with 50 ng/μl of a coelomocyte co-injection marker (*unc-122p::GFP*) and 30 ng/μl of 1 kb DNA ladder. For the *tax-4* gene, the injection mixtures were prepared using 40 ng/μl of *tax-4* expression vectors, 50 ng/μl of a coelomocyte co-injection marker, and 10 ng/μl of 1 kb DNA ladder. The rest of the expression constructs were injected at 50 ng/μl together with 50 ng/μl of a coelomocyte marker.

### CRISPR/Cas9 genome editing

Genes were disrupted using CRISPR/Cas9-mediated genome editing as described (*Dokshin et al., 2018*; *Ghanta and Mello, 2020*). The strategy involved the utilization of homology-directed insertion of custom-designed single-strand DNA template (ssODN), which had two homology arms of 35 bp flanking the targeted PAM site. Between two homology arms, a short sequence containing a unique restriction enzyme cutting site as well as in-frame and out-of-frame stop codons was included. The insertion of the ssODN template would delete 16 bases of coding sequence, ensuring the proper gene disruption. To prepare the injection cocktail, 0.5 μl of Cas9 protein (IDT) was mixed with 5 μl of 0.4 μg/μl tracrRNA (IDT, United States) and 2.8 μl of 0.4 μg/μl crRNA (IDT). The mixture was incubated at 37 °C for at least 10 min before 2.2 μl of 1 μg/μl ssODN (or 500 ng dsDNA) and 2 μl of 0.6 μg/μl *rol-6* co-injection marker were added. Nuclease-free water was used to bring the final volume to 20 μl. The injection mixture was centrifuged for 2 min before use.

### Behavioral assays

$H_2S$-evoked locomotion activity was monitored as described previously (*Laurent et al., 2015*; *Zhao et al., 2022*). Briefly, OP50 bacteria obtained from the *Caenorhabditis* Genetics Center were seeded on the assay plates 16 hr before use. The border of the bacterial lawn was removed using a PDMS stamp. Day-1 adult hermaphrodites were used in all assays unless otherwise specified. For each assay, 25–30 day-1 adult animals were picked onto assay plates, allowed to settle down for 15 min, and subsequently sealed within microfluidic chambers. A syringe pump (PHD 2000, Harvard Apparatus)

was employed to deliver gas mixtures into the microfluidic chamber at a constant flow of 3 ml/min. The rapid switch between different gas mixtures was controlled by Teflon valves coupled with Valve-Bank Perfusion Controller (AutoMate Scientific). The locomotory activity at different gas mixtures was monitored using a high-resolution camera (FLIR) mounted on a Zeiss Stemi 508 microscope. Videos were captured at a rate of 2 frames per second, starting with a 2-min recording in 7% $O_2$, followed by 10 or 11 min in $H_2S$, and ending with 7% $O_2$. For the long-term recording in $H_2S$, video capture commenced with a 2-min period in 7% $O_2$, followed by a duration of 148 min in 150 ppm $H_2S$, and concluded with a final 2-min interval in 7% $O_2$. For each strain or condition, three to four replicates were performed.

Near-UV light experiments were conducted to assess whether animals remained responsive following a 30-min incubation in 150 ppm $H_2S$ balanced with 7% $O_2$. For each experiment, 6-min videos were recorded. The first 2 min captured the baseline locomotion of the animals under white light, followed by 2 min of exposure to near-UV light (435 nm, 0.7 mW/mm$^2$), and concluded with 2 min of white light. 3–4 replicates were performed, with approximately 15 animals recorded per replicate.

The $H_2S$ gradient experiment was performed in a PDMS chamber connected to a pump delivering gas at 0.5 ml/min. To establish a gradient ranging from 150 ppm to 0 ppm $H_2S$, we pumped 150 ppm $H_2S$ balanced with 7% $O_2$ into one side and 7% $O_2$ into the opposite side. After 25 min, the number of animals in each of the five sections was counted (*Figure 1G*). Four replicates were performed, each using 30 day-1 adult animals.

To assess the effects of rotenone on $H_2S$-evoked locomotory response, day-1 adult animals were subjected to 10 μM rotenone for 10 min, 30 min, 1 hr, and 2 hr on the drug-containing plates. Subsequently, the rotenone-treated animals were assayed on the standard assay plates without the drug. To explore the impact of ferric ammonium citrate (FAC) (F5879, Sigma-Aldrich) and 2,2'-Bipyridyl (BP) (D216305, Sigma-Aldrich) on behavioral response to $H_2S$, L4 animals were exposed to 100 μM FAC or 5 mg/ml BP for 16 hr in the presence of bacterial food. The FAC or BP-treated animals were assayed on FAC or BP-containing plates, respectively.

Optogenetic experiments were performed as previously outlined (*Zhao et al., 2022*). Transgenic L4 animals were exposed to 100 μM all-trans retinal (ATR; R2500, Sigma) in the dark for 16 hr. The ATR-fed animals were assayed under continuous illumination of 70 μW/mm$^2$ blue light, which was switched on at the start of recording at 7% $O_2$. Blue light was emitted from an ultra-high-power LED lamp (UHP-MIC-LED-460, Prizmatix), and the light intensity was determined using a PM50 Optical Power Meter (ThorLabs). To minimize the effect of transmitted light on the transgenic animals, a long-pass optical filter was used to eliminate the lights with short wavelengths during picking. All videos of behavioral analysis were analyzed using a home-made MATLAB (RRID:SCR_001622) program Zentracker (https://github.com/wormtracker/zentracker; *wormtracker, 2015*). For all behavior analyses, at least three assays were performed for each strain with more than 80 worms in total.

## Ca²⁺ imaging

L4 animals expressing the GCaMP6s sensor in ASJ neurons were picked 24 hr before imaging. The agarose pads were prepared 1 hr before the experiment. Animals were incubated in 15 μl of 10 mM levamisole for 40 min prior to imaging. Between 15 and 20 animals were immobilized using 10 mM levamisole diluted in a concentrated OP50 bacterial suspension and mounted on a 2% agarose pad in M9 buffer on a glass slide. The animals were then covered with a microfluidic chamber. Imaging lasted for 16 min. $O_2$ (7%) was pumped into the microfluidic chamber during the first and last 4 min. From min 4 to 12, a mixture of 7% $O_2$ and 150 ppm $H_2S$ was pumped.

Imaging was performed on a Nikon AZ100 microscope equipped with a TwinCam adapter (Cairn Research, UK), fitted with two DMK 33 U monochrome cameras (The Imaging Source, Germany). A 2 x AZ Plan Fluor objective with 1 x zoom was used, with an exposure time of 300 ms. Excitation of GCaMP and RFP was provided by a CoolLED pE-300 (CoolLED, UK). The TwinCam adapter was mounted with emission filters for green fluorescent protein (510/20 nm) and red fluorescent protein (590/35 nm), along with a DC/T510LPXRXTUf2 dichroic mirror. Imaging data were analyzed using Neuron Analyzer, a custom-written MATLAB (RRID:SCR_001622) program for processing image stacks (code available at https://github.com/neuronanalyser/neuronanalyser; *neuronanalyser, 2015*).

## RNA extraction and sequencing

To obtain RNA samples for RNA-seq, 30 young adult animals were picked to each fresh plate and allowed to lay eggs for 2 hr, after which the adult animals were removed and eggs were allowed to develop into day-1 adults. Day-1 animals were subsequently challenged with $H_2S$ for three different time periods (1 hr, 2 hr, and 12 hr). For each time period, five plates of day-1 animals were exposed to 50 ppm $H_2S$, 150 ppm $H_2S$, or 7% $O_2$ as a control group. Subsequently, the animals were immediately collected and rinsed three times with M9 buffer. After washing, the worm pellet was frozen in liquid nitrogen. Animals were homogenized using Bullet Blender (Next Advance) in the presence of Qiazol Lysis Reagent and 0.5 mm Zirconia beads at 4 °C. RNA was prepared using RNeasy Plus Universal Mini Kit (QIAGEN). Six independent RNA samples were prepared for each condition. The 2100 Bioanalyzer instrument (Agilent) was used for the RNA quality control with a microfluidic chip specific for RNA (Agilent RNA 6000 Nano Kit). Then 1 µg of qualified RNA samples in RNase-free ddH2O was used for library preparation. The library was constructed and sequenced by Novogene.

## RNA-seq analysis

The RNA-seq data were aligned using STAR v2.7.9a (*Dobin et al., 2013*; RRID:SCR_004463) and gene expression counts extracted by featureCounts v2.0.3 (*Liao et al., 2014*; RRID:SCR_012919). Differential expression was calculated using DESeq2 (*Love et al., 2014*; RRID:SCR_000154). GO analysis was performed using the EnrichR R package (*Kuleshov et al., 2016*) on genes having p.adj<1e-10. For brevity, several highly similar GO categories were manually omitted (full EnrichR output available at GitHub repository).

## Statistics details

Stage-synchronized worms were randomly assigned into experimental and control groups for each genotype. Experiments were conducted with blinding to genotype and treatment. Sample sizes in behavioral assays are determined based on established methods and standardized procedures in *C. elegans* research. All data were included in the analyses. Locomotory speeds between genotypes were compared using in MATLAB (RRID:SCR_001622) scripts.

## Materials availability

All reagents generated in this study, including strains and plasmids, will be made available upon request.

## Acknowledgements

We thank the Caenorhabditis Genetics Center (funded by NIH Office of Research Infrastructure Programs P40 OD010440) and the National BioResources Project Japan for strains.

## Additional information

### Funding

| Funder | Grant reference number | Author |
| --- | --- | --- |
| Vetenskapsrådet | 2021-06602 | Johan Henriksson |
| Fonds De La Recherche Scientifique - FNRS | 40020876 | Patrick Laurent |
| Vetenskapsrådet | 2018-02216 | Changchun Chen |
| Vetenskapsrådet | 2024-04141 | Changchun Chen |
| European Research Council | 802653 | Changchun Chen |
| Fonds De La Recherche Scientifique - FNRS | 40038114 | Clementine Deleuze |

| Funder | Grant reference number | Author |
|--------|------------------------|--------|

The funders had no role in study design, data collection and interpretation, or the decision to submit the work for publication.

## Author contributions

Longjun Pu, Lina Zhao, Jing Wang, Lars Nilsson, Data curation, Formal analysis, Validation, Investigation, Visualization, Methodology; Clementine Deleuze, Data curation, Formal analysis, Supervision, Validation, Investigation, Visualization, Methodology; Johan Henriksson, Data curation, Formal analysis, Validation, Visualization, Methodology; Patrick Laurent, Conceptualization, Resources, Formal analysis, Supervision, Funding acquisition, Validation, Investigation, Visualization, Methodology, Writing – review and editing; Changchun Chen, Conceptualization, Resources, Data curation, Formal analysis, Supervision, Funding acquisition, Validation, Investigation, Visualization, Methodology, Writing – original draft, Project administration, Writing – review and editing

## Author ORCIDs

Longjun Pu ⓘ https://orcid.org/0000-0002-0239-8732
Clementine Deleuze ⓘ https://orcid.org/0009-0008-9581-9575
Johan Henriksson ⓘ https://orcid.org/0000-0002-7745-2844
Patrick Laurent ⓘ https://orcid.org/0000-0001-5360-5597
Changchun Chen ⓘ https://orcid.org/0000-0003-2233-8996

Reviewer #3 (Public review): https://doi.org/10.7554/eLife.92964.4.sa1
Reviewer #4 (Public review): https://doi.org/10.7554/eLife.92964.4.sa2
Author response https://doi.org/10.7554/eLife.92964.4.sa3

# Additional files

## Supplementary files

Supplementary file 1. $H_2S$-evoked behavioral responses in mutants from the candidate gene screen.

Supplementary file 2. RNA-seq analysis of gene expression responses to 50 or 150 ppm $H_2S$ at different time points.

Supplementary file 3. List of strains used in this study.

Supplementary file 4. List of reagents and primers used in this study.

MDAR checklist

## Data availability

Raw sequencing data has been deposited at ArrayExpress (RRID:SCR_002964) # E-MTAB-13296. All data generated or analyzed during this study are included in the manuscript and supporting files. The R code is available at https://github.com/henriksson-lab/ce_h2s_tc (copy archived at *Henriksson, 2025*).

The following dataset was generated:

| Author(s) | Year | Dataset title | Dataset URL | Database and Identifier |
|-----------|------|---------------|-------------|--------------------------|
| Henriksson J | 2025 | *C. elegans* H2S RNA-seq time course | https://www.ebi.ac.uk/biostudies/ArrayExpress/studies/E-MTAB-13296?query=E-MTAB-13296 | EBI ArrayExpress, E-MTAB-13296 |

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
