## [Editor Report · eLife Assessment]

These **valuable** studies explore the consequences of exposure to the toxin hydrogen sulfide (H2S) on the behavior and physiology of *C. elegans*. The work finds that behavioral changes evoked by H2S exposure are modulated by several regulatory pathways known to influence chemosensory-evoked locomotor behavior, but there is **incomplete** data to support the authors' claim of comprehensive mechanistic insight into the consequences of H2S exposure. Nevertheless, the findings may be informative for those studying organismal stress responses and the effects of mitochondrial ROS on behavior and physiology.

---

## [Referee Report · Reviewer #3 (Public review)]

Summary:

The manuscript explores behavioral responses of *C. elegans* to hydrogen sulfide, which is known to exert remarkable effects on animal physiology in a range of contexts. The possibility of genetic and precise neuronal dissection of responses to H2S motivates the study of responses in *C. elegans*.

The authors have followed up observations in the initial version of the manuscript, and their data do not support the direct sensing of H2S by the ASJ neurons or other sensory neurons. Genetic and parallel analysis of O2 and CO2 responsive pathways do not reveal further insights regarding potential mechanisms underlying H2S sensing. Gene expression analysis extends prior work. Finally, the authors have examined how H2S-evoked locomotory behavioral responses are affected in mutants with altered stress and detoxification response to H2S, most notably hif-1 and egl-9. These data, while examining locomotion, are more suggestive that observed effects on animal locomotion are secondary to altered organismal toxicity as opposed to specific behavioral responedse

Overall, the manuscript provides a wide range of preliminary observations of genetic interactions that may influence locomotory responses to H2S, but mechanistic insight or a synthesis of disparate data is lacking.

---

## [Referee Report · Reviewer #4 (Public review)]

Summary:

The authors establish a behavioral paradigm for avoidance of H2S and conduct a large candidate screen to identify genetic requirements. They follow up by genetically dissecting a large number of implicated pathways - insulin, TGF-beta, oxygen/HIF-1, and mitochondrial ROS, which have varied effects on H2S avoidance. They additionally assay whole-animal gene expression changes induced by varying concentrations and durations of H2S exposure.

Strengths:

The implicated pathways are tested extensively through mutants of multiple pathway molecules. The authors address previous reviewer concerns by directly testing the ability of ASJ to respond to H2S via calcium imaging. This allows the authors to revise their previous conclusion and determine that ASJ does not directly respond to H2S and likely does not initiate the behavioral response. Extensive experiments manipulating the mitochondrial ETC and ROS support the authors' revised model that mitochondrial toxicity is the major driver of H2S avoidance.

It seems possible that HIF-1 and SKN-1 signaling directly modulate ROS toxicity while ASJ neurons and the oxygen sensing circuit could modulate the avoidance behavior. How this neuronal interaction happens remains unknown.

---

## [Author Response]

The following is the authors’ response to the previous reviews.

**Reviewer #3 (Public review):**
Summary:The manuscript explores behavioral responses of *C. elegans* to hydrogen sulfide, which is known to exert remarkable effects on animal physiology in a range of contexts. The possibility of genetic and precise neuronal dissection of responses to H2S motivates the study of responses in *C. elegans*. The revised manuscript does not seem to have significantly addressed what was lacking in the initial version.The authors have added further characterization of possible ASJ sensing of H2S by calcium imaging but ASJ does not appear to be directly involved. Genetic and parallel analysis of O2 and CO2 responsive pathways do not reveal further insights regarding potential mechanisms underlying H2S sensing. Gene expression analysis extends prior work. Finally, the authors have examined how H2S-evoked locomotory behavioral responses are affected in mutants with altered stress and detoxification response to H2S, most notably hif-1 and egl-9. These data, while examining locomotion, are more suggestive that observed effects on animal locomotion are secondary to altered organismal toxicity as opposed to specific behavioral responedseOverall, the manuscript provides a wide range of intriguing observations, but mechanistic insight or a synthesis of disparate data is lacking.

We thank the reviewer for the valuable feedback. We agree that while our investigation provides broad coverage, it does not fully resolve the mechanisms of H_2_S perception. As both reviewers noted, the avoidance response to high levels of H_2_S is most likely driven by its toxicity, particularly at the level of mitochondria, rather than by direct perception of H_2_S. We also favor this model and have revised the results and discussion to highlight this interpretation, while acknowledging that other mechanisms cannot be excluded (main changes lines 387-402 and 535-547).

Building on this view, our observations point toward mitochondrial ROS transients as the trigger for H_2_S avoidance. First, toxic levels of H_2_S are known to promote ROS production (1). Second, similar to acute H_2_S, brief exposure to rotenone, an ETC complex I inhibitor that rapidly generates mitochondrial ROS, triggers locomotory responses (Figure 7E) (Lines 393-396). Third, regardless of duration, rotenone exposure inhibits H_2_S-evoked avoidance (Figure 7E) (Lines 389-391), likely by preventing or dampening H_2_S-evoked mitochondrial ROS bursts when ETC function is impaired and ROS is already high. Notably, animals subjected to prolonged rotenone exposure, ETC mutants, and quintuple sod mutants, each experiencing chronically high ROS levels, fail to respond to H_2_S and display reduced locomotory activity, presumably due to ROS toxicity and/or activation of stress-adaptive mechanisms (Figure 7).

Consistent with the activation of stress-responsive pathways, H_2_S exposure alters expression of genes controlled by SKN-1 and HIF-1 signaling. Both pathways are ROS-sensitive and promote adaptation to chronic ROS production (2-4). Their activation, as in egl-9, render these animals insensitive to H_2_S-evoked ROS transients (Figure 5B) (Lines 303-305). Conversely, mutants defective in these adaptive pathways, such as hif-1, still show initial locomotory responses to H_2_S, but rapidly lose activity during prolonged H_2_S exposure (Figure 5D) (Lines 318-319). These observations suggest that HIF-1 pathway is dispensable for initiating the response to H_2_S evoked ROS transients, but essential for protecting against ROS toxicity.

In this context, the neural circuit we examined, such as ASJ neurons, is not directly involved in H_2_S perception (Line 165-169 and 448-457). Instead, it likely modulates a circuit that is responsive to ROS toxicity. This circuit is also influenced by ambient O_2_ levels, the state of O_2_ sensing circuit, and nutrient status, in a manner reminiscent of the CO_2_ responses (5, 6).

**Reviewer #4 (Public review):**
Summary:The authors establish a behavioral paradigm for avoidance of H2S and conduct a large candidate screen to identify genetic requirements. They follow up by genetically dissecting a large number of implicated pathways - insulin, TGF-beta, oxygen/HIF-1, and mitochondrial ROS, which have varied effects on H2S avoidance. They additionally assay whole-animal gene expression changes induced by varying concentrations and durations of H2S exposure.Strengths:The implicated pathways are tested extensively through mutants of multiple pathway molecules. The authors address previous reviewer concerns by directly testing the ability of ASJ to respond to H2S via calcium imaging. This allows the authors to revise their previous conclusion and determine that ASJ does not directly respond to H2S and likely does not initiate the behavioral response.

We thank the reviewer for the supportive comments.

Weaknesses:Despite the authors focus on acute perception of H2S, I don't think the experiments tell us much about perception. I think they indicate pathways that modulate the behavior when disrupted, especially because most manipulations used broadly affect physiology on long timescales. For instance, genetic manipulation of ASJ signaling, oxygen sensing, HIF-1 signaling, mitochondrial function, as well as starvation are all expected to constitutively alter animal physiology, which could indirectly modulate responses to H2S. The authors rule out effects on general locomotion in some cases, but other physiological changes could relatively specifically modulate the H2S response without being involved in its perception.I am actually not convinced that H2S is directly perceived by the *C. elegans* nervous system at all. As far as I can tell, the avoidance behavior could be a response to H2S-induced tissue damage rather than the gas itself.

We thank the reviewer for the valuable insights, and fully agree that the H_2_S may not be directly perceived by *C. elegans*. Please see detailed responses below.

**Reviewer #4 (Recommendations for the authors):**
The clarity of the paper is improved in this version. My main issue has to do with "perception" of H2S. At times the authors suggest that hydrogen sulfide should be perceived by a neural circuit ("we did not specifically identify the neural circuit mediating H2S signaling"), while at other times they discuss the possibility that it is not directly perceived neuronally ("Supporting the idea that acute mitochondrial ROS generation initiates avoidance of high H2S levels,"). The authors should clearly state their model for H2S perception. Do they think there is a receptor and sensory neuron for H2S (not identified in this paper)? If not, what does it mean for there to be a neural circuit mediating the response? To me, it looks more like what is being "perceived" by a neural circuit is ROS-induced toxicity, not H2S itself.To drill down on direct modulation of acute perception, are any of the pathway manipulations used in this paper performed on the timescale of perception? Rotenone for 10 mins is close to that timescale, and in fact it increases speed independently of H2S, consistent with ROSinduced toxicity, not H2S being the signal that induces the behavior. Optogenetic activation of RMG could also be on the acute timescale. Can the authors clarify for how long blue light was on the worms before the start of the assay? Or was it turned on at the same time as video acquisition commenced? This could be evidence that RMG acutely modulates this behavioral response.I feel that the ASJ calcium imaging data should be in the main figure given its importance in revising the original model.

We thank the reviewer for the valuable advice.

As suggested, ASJ calcium imaging data are displayed in the main figure (Figure 2I) (Line 167).

As both reviewers noted, our initial presentation was not sufficiently clear regarding the mechanism underlying H_2_S avoidance. We agree with the reviewer that H_2_S avoidance is unlikely mediated by direct perception via a H_2_S-specific receptor, but likely arises from acute mitochondrial dysfunction and ROS generation.

ROS

In line with the reviewer’s perspective, our observations point toward mitochondrial ROS transients as the trigger for H_2_S avoidance. First, toxic levels of H_2_S are known to promote ROS production (1). Second, similar to acute H_2_S, brief exposure to rotenone, an ETC complex I inhibitor that rapidly generates mitochondrial ROS, triggers locomotory responses (Figure 7E) (Lines 393-396). Third, regardless of duration, rotenone exposure inhibits H_2_S-evoked avoidance (Figure 7E) (Lines 389-391), likely by preventing or dampening H_2_S-evoked mitochondrial ROS bursts when ETC function is impaired and ROS is already high. Notably, animals subjected to prolonged rotenone exposure, ETC mutants, and quintuple sod mutants, each experiencing chronically high ROS levels, fail to respond to H_2_S and display reduced locomotory activity, presumably due to ROS toxicity and/or activation of stress-adaptive mechanisms (Figure 7). We revised the Results and Discussion to present the model more consistently (main changes lines 387-402 and 535-547).

Consistent with the activation of stress-responsive pathways, H_2_S exposure alters expression of genes controlled by SKN-1 and HIF-1 signaling. Both pathways are ROS-sensitive and promote adaptation to chronic ROS production (2-4). Their activation, as in egl-9, render these animals insensitive to H_2_S-evoked ROS transients (Figure 5B) (Lines 303-305). Conversely, mutants defective in these adaptive pathways, such as hif-1, still show initial locomotory responses to H_2_S, but rapidly lose activity during prolonged H_2_S exposure (Figure 5D) (Lines 318-319). These observations suggest that HIF-1 pathway is dispensable for initiating the response to H_2_ Sevoked ROS transients, but essential for protecting against ROS toxicity.

ASJ neurons

ASJ neurons and DAF-11 signaling are required for H_2_S-evoked behavioral responses. However, ASJ does not exhibit an H_2_S-evoked calcium transient. It suggests that ASJ neurons do not directly detect H_2_S (Line 165-169 and 448-457), but likely modulate the circuit responsive to ROS toxicity. This circuit can also be modulated by ambient O_2_ levels, the state of O_2_ sensing circuit, and nutrient status, in a manner reminiscent of the CO_2_ responses (5, 6).

O_2_ sensing circuit

Consistent with the reviewer’s view, we favor the model that H_2_S avoidance is likely induced by ROS transients. We believe that the state of O_2_ sensing circuit, similar to ASJ neurons, modulates the neural circuit that is responsive to H_2_S-evoked ROS toxicity. This circuit is inhibited as long as O_2_ sensing circuit is active. In the RMG optogenetic experiment, channelrhodopsin was photo-stimulated as soon as the assay was initiated at 7% O_2_ (Methods Lines 633-634 and Figure legend Lines 1177-1178), therefore RMG remained active throughout the assay including at 7% O_2_. Our interpretation is that RMG activation inhibits this ROSresponsive circuit and H_2_S avoidance. However, these observations do not resolve if H_2_S is acutely and directly perceived. The modulation of H_2_S response by O_2_ circuit was discussed between Lines 437-447.

References

(1) J. Jia et al., SQR mediates therapeutic effects of H(2)S by targeting mitochondrial electron transport to induce mitochondrial uncoupling. Sci Adv 6, eaaz5752 (2020).

(2) S. J. Lee, A. B. Hwang, C. Kenyon, Inhibition of Respiration Extends *C. elegans* Life Span via Reactive Oxygen Species that Increase HIF-1 Activity. Current Biology 20, 2131-2136 (2010).

(3) C. Lennicke, H. M. Cocheme, Redox metabolism: ROS as specific molecular regulators of cell signaling and function. Mol Cell 81, 3691-3707 (2021).

(4) D. A. Patten, M. Germain, M. A. Kelly, R. S. Slack, Reactive oxygen species: stuck in the middle of neurodegeneration. J Alzheimers Dis 20 Suppl 2, S357-367 (2010).

(5) A. J. Bretscher, K. E. Busch, M. de Bono, A carbon dioxide avoidance behavior is integrated with responses to ambient oxygen and food in *Caenorhabditis elegans*. Proc Natl Acad Sci U S A 105, 8044-8049 (2008).

(6) E. A. Hallem, P. W. Sternberg, Acute carbon dioxide avoidance in *Caenorhabditis elegans*. Proc Natl Acad Sci U S A 105, 8038-8043 (2008).